# Comparative transcriptomics coupled to developmental grading via transgenic zebrafish reporter strains identifies conserved features in neutrophil maturation

Stefanie Kirchberger [1,10,11] ✉, Mohamed R. Shoeb [1,10], Daria Lazic[1], Andrea Wenninger-Weinzierl[1], Kristin Fischer [1], Lisa E. Shaw [2], Filomena Nogueira [1,3,4], Fikret Rifatbegovic [1], Eva Bozsaky [1], Ruth Ladenstein [1], Bernd Bodenmiller [5,6], Thomas Lion[1,3,7], David Traver [8], Matthias Farlik [2], Christian Schöfer [9], Sabine Taschner-Mandl [1], Florian Halbritter [1,11] ✉ & Martin Distel [1,11] ✉

Neutrophils are evolutionarily conserved innate immune cells playing pivotal roles in host defense. Zebrafish models have contributed substantially to our understanding of neutrophil functions but similarities to human neutrophil maturation have not been systematically characterized, which limits their applicability to studying human disease. Here we show, by generating and analysing transgenic zebrafish strains representing distinct neutrophil differentiation stages, a high-resolution transcriptional profile of neutrophil maturation. We link gene expression at each stage to characteristic transcription factors, including C/ebp-β, which is important for late neutrophil maturation. Cross-species comparison of zebrafish, mouse, and human samples confirms high molecular similarity of immature stages and discriminates zebrafish-specific from pan-species gene signatures. Applying the pan-species neutrophil maturation signature to RNA-sequencing data from human neuroblastoma patients reveals association between metastatic tumor cell infiltration in the bone marrow and an overall increase in mature neutrophils. Our detailed neutrophil maturation atlas thus provides a valuable resource for studying neutrophil function at different stages across species in health and disease.

Neutrophils are the most abundant immune cell population in humans and the first responders to injury and infection[1,2]. In mammals, neutrophils mature in the bone marrow (BM), during the final steps of a cascade where hematopoietic stem cells differentiate through a granulocyte-monocyte progenitor towards neutrophils[3,4]. Characteristic granules form throughout development from the promyelocyte to segmented neutrophil stage[5–7]. Transcription factors of the C/EBP family are key for the expression of granule enzymes[8,9]. Surface marker expression, nuclear morphology, and granule content have long been used to define neutrophil maturation stages, but recent studies using single-cell RNA sequencing (scRNA-seq) and proteome analysis (CyTOF) have questioned this classification scheme instead suggesting a sequence of continuous transcriptional stages[10–15].

Functional diversity of neutrophils at different maturation grades is still understudied[16,17]. A potential role for different maturation states becomes apparent in cancer, where immature neutrophils often accumulate in blood and tumors, and can have altered effector functions such as reduced phagocytosis, ROS production, NETosis, granularity, chemokine receptor expression and increased suppressive functions[6,16,18]. Neutrophils are now known to be involved in almost every stage of cancer such as tumor initiation by ROS production, growth, angiogenesis, and the conditioning of the pre-metastatic niche[6,17]. Different tumor-associated neutrophil (TAN) populations polarized by TGF-β or IFN-β towards pro- or anti-tumor roles have been observed. Interestingly, their opposing roles in cancer progression have been linked to different maturation stages and densities[18–20].

Zebrafish are a versatile model to study neutrophil functions in infection and tumorigenesis thanks to the availability of many fluorescent transgenic lines and the prospect of intravital imaging[21–23]. Hematopoiesis in zebrafish occurs in the kidney marrow, where all major immune cell types known from humans are present and are generally considered evolutionarily conserved[24,25]. However, to date the transcriptional states of zebrafish neutrophils during maturation and associated functions in vivo have not been mapped to their human equivalents, thus limiting cross-species comparisons. Comparative transcriptomic studies have been hampered by underrepresentation of neutrophils in many human datasets due to technical difficulties and by a lack of suitable zebrafish lines allowing effective sorting of neutrophils comprising multiple maturation stages[26].

Here, we establish *Tg(lysC:CFPNTR)*[vi002]/ *Tg(BACmmp9:Citrine-CAAX)*[vi003] double transgenic zebrafish, which allow us to distinguish immature from mature neutrophils in vivo, visualize their interactions with bacteria and tumor cells, and isolate maturing neutrophil populations for morphological and transcriptional analysis. We use scRNA-seq to catalog transcriptional changes during maturation and to identify critical transcription factors, including C/ebp-β. Finally, cross-species comparison enables the definition of a conserved gene signature, which we apply to analyze bulk human tumor transcriptome data and to correlate neutrophil maturation stage with BM metastasis.

## Results

### The Mmp9 transgene identifies mature neutrophils in zebrafish

In order to distinguish immature from mature neutrophils in vivo, we generated transgenic zebrafish expressing membrane-directed Citrine under the control of regulatory elements for *mmp9* (*Tg(BACmmp9:Citrine-CAAX)*[vi003]), a tertiary granule protein in mammalian mature neutrophils and expressed in zebrafish mature heterophils[27,28]. *Tg(BACmmp9:Citrine-CAAX)*[vi003] zebrafish embryos/larvae report *mmp9* transcription from 2 days post fertilization (dpf) in the epithelia of the tail fin and the distal gut (Supplementary Fig. 1a). Additionally, a dotted pattern became apparent along the head, yolk, and in the caudal hematopoietic tissue (CHT) suggesting expression in a leukocyte population. We confirmed that Citrine fluorescence specifically reports *mmp9* expression by detecting *mmp9* RNA in FACS-purified Citrine⁺ but not in Citrine⁻ cells (Supplementary Fig. 1b).

To examine the identity of Mmp9⁺ leukocytes, mmp9:Citrine fish were crossed with fish double-transgenic for myeloid markers *Tg(lysC:CFP-NTR)*[vi002] (labeling neutrophils), and *Tg(mpeg:mCherry)*[gl23] (macrophages)[29]. Live imaging of triple-transgenic larvae at 3 and 5 dpf confirmed the existence of both Mmp9⁻ and Mmp9⁺ subpopulations of *lysC:CFP*⁺ neutrophils with a more pronounced co-localization in the head than in the caudal hematopoietic tissue region (Fig. 1a). Mmp9⁺LysC⁺ cells had low mpeg:mCherry expression, while bona-fide macrophages were highly (2.5-fold higher signal intensity) positive for mpeg:mCherry and few of the latter showed detectable *mmp9* expression (Fig. 1b and Supplementary Fig. 1c). LysC⁺Mpeg⁺Mmp9⁺ triple-positive neutrophils were enriched in the head region at 3 dpf (mean = 36%; range = 24.3–52.3%) as well as in the head and tail at 5 dpf

(mean = 37%; range = 29–46.8%; and 32%, range = 20.5–40%, respectively), whereas the CHT population decreased from 3 to 5 dpf (from mean = 20%, range = 11.1–30.3%, to 13%, range = 10–17.6%) (Supplementary Fig. 1c). Pre-dominant localization of triple-positive neutrophils in the head points towards an already more progressed maturation stage of RBI-derived head neutrophils compared to HSC-derived CHT neutrophils. Interestingly, a recent publication also described transcriptional differences between these two neutrophil populations[30].

Complementary analysis by flow cytometry revealed that 15.4% (mean; range: 12.4–17.1%) of myeloid cells were expressing Mmp9; of those 67% (mean; range 61.8–69.5%; Q2) belonged to the LysC⁺Mpeg⁺ neutrophil population (Fig. 1c, Supplementary Fig. 2a). Fewer Citrine-positive cells were part of the LysC⁺Mpeg⁻ neutrophil (mean= 29.06%; range: 25.8–34.8% Q1) and Mpeg⁺LysC⁻ macrophage (mean = 3.9%; range 2.8–5.8%; Q3) populations. Mmp9 expression was restricted to a subpopulation of cells stained with Sudan Black, a lipophilic dye labeling granules present in maturing neutrophils (Supplementary Fig. 1d)[31]. Furthermore, Mmp9⁺ cells showed an increased side scatter (SSC) compared to Mmp9⁻ cells indicating a higher granularity and intracellular complexity, which suggests a mature phenotype (Fig. 1d).

Time-lapse imaging of cellular behavior after wounding showed that both, LysC⁺Mmp9⁺ and LysC⁺Mmp9⁻ neutrophils had a similar round morphology and moved to and from the wound quickly with amoeboid motility as described for neutrophils[32] (Supplementary Movie 1). In contrast, Mpeg⁺ cells showed the protrusions typical of macrophages and stayed close to the wound edge.

In adult zebrafish LysC⁺Mmp9⁺neutrophils were detectable in the whole kidney marrow (WKM), the primary hematopoietic organ of teleost fish, as well as in the spleen and blood (Fig. 1e). We investigated morphology and maturation grade of FACS-sorted LysC⁺Mmp9⁺ and LysC⁺Mmp9⁻ populations from WKM on stained cytospins and by electron microscopy (Fig. 1f, g, Supplementary Fig. 1e–i). Only 3% (mean, range = 0–9%) of Mmp9⁻ cells contained segmented nuclei compared to 23% (mean, range = 3–47%) of the Mmp9⁺ fraction, indicating an advanced maturation grade of the latter (Fig. 1f). LysC⁺Mmp9^HI neutrophils were also of smaller cell size than Mmp9⁻ cells another parameter for a more differentiated stage (Supplementary Fig. 1g). The ultrastructure of LysC⁺Mmp9^HI cells revealed more developed cigar-shaped granules, while granules in Mmp9⁻ cells were generally rounder (Fig. 1g, Supplementary Fig. 1i). Collectively, our data show that Mmp9 is a suitable marker for mature neutrophils in zebrafish consistent with previous data for human cells[33].

### Mmp9⁺ neutrophils are highly phagocytic and rapidly recruited to wounds

Neutrophils are rapidly recruited to sites of injury through chemotactic cues[34]. When zebrafish larvae were wounded at the ventral fin, Mmp9^HI neutrophil tracks came closer to the wound (mean = 8.3 μm HI, 45.8 μm NO, 43 μm INT; *n* = 6) and showed a lower linearity of movement (mean = 0.23 HI, 0.42 NO, 0.39 INT; *n* = 6) than Mmp9^INT or ^NO cells resulting from a meandering movement in the wound area, while their speed was similar (mean = 0.055 μm/s HI, 0.045 μm/s NO, 0.053 μm/s INT) (Fig. 2a–d). The phagocytic capacity of neutrophils increases with their maturation grade[35]. We performed in vivo phagocytosis assays by injecting live mCherry-labeled *E. coli* into 2 dpf *Tg(lysC:CFPNTR)*[vi002]/ *Tg(BACmmp9:Citrine-CAAX)*[vi003] larvae (Fig. 2e–g; Supplementary Fig. 2b). Both neutrophil subpopulations, LysC⁺Mmp9⁻ and LysC⁺Mmp9⁺, were able to ingest bacteria (Fig. 2e) and to produce reactive oxygen species (Supplementary Fig. 2c). Quantification of *E. coli* uptake as seen by overlap of mCherry (*E. coli*) and CFP (neutrophil) fluorescence by flow cytometry showed a significantly higher percentage of Mmp9⁺ neutrophils with bacterial cargo compared to Mmp9⁻ neutrophils (mean = 23.8% and 10.7%, respectively; *n* = 5; paired *t* test, *P* = 0.007) (Fig. 2f; Supplementary Fig. 2b). Furthermore, Mmp9⁺

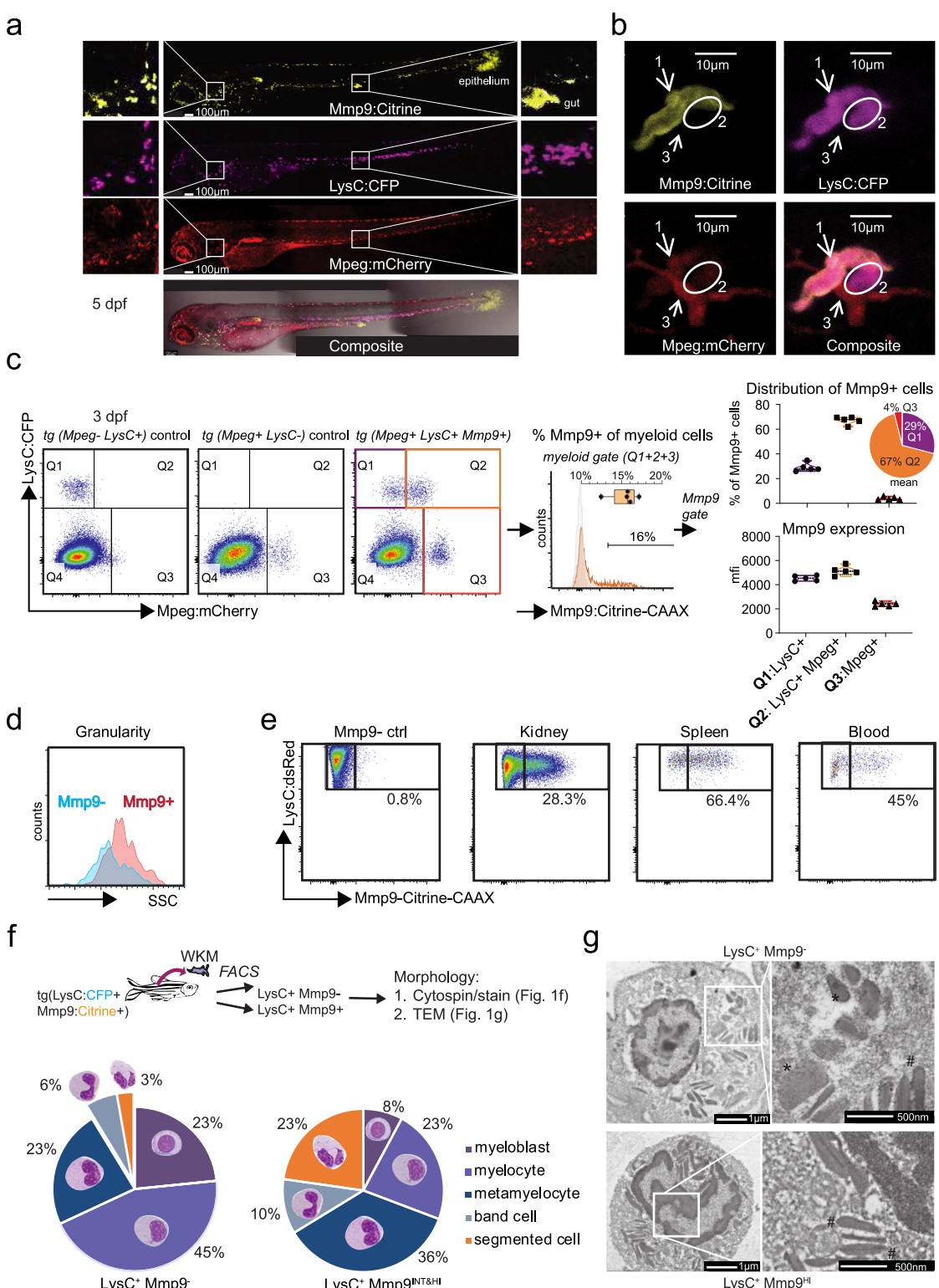

neutrophils were more efficiently recruited to sites of bacterial infection (67.3% Mmp9⁺ vs. 32.7% Mmp9⁻ of all LysC⁺ neutrophils at infection site; n = 33; paired t test; P = 0.002) (Fig. 2g) and the overall frequency of Mmp9⁺ cells was increased during infection (mean = 56.4% Mmp9⁺ in infected and 39.2% Mmp9⁺ in PBS treated, n = 5; paired t test; P = 0.009) (Supplementary Fig. 2d, e).

Next, we examined neutrophils in a model of pre-neoplastic melanoma (Et(kita:GAL4)^hzm1 x Tg(HRAS_G12V:UAS:CFP)^vi004)[36]. As during infection, we found an increase in Mmp9⁺ cells (mean 46.0% Mmp9⁺ in

the presence of HRAS^G12V versus 37.4% Mmp9⁺ in controls; n = 5; paired t test; P = 0.005) (Supplementary Fig. 2f). Live imaging in transparent zebrafish larvae showed that some Mmp9⁺ cells stayed in contact with transformed cells by thin tethers (Fig. 2h) and others interacted over a long period (Supplementary Movie 2), consistent with previous data for LysC⁺ neutrophils[37]. We found that some neutrophils got into close contact, spread out, and crawled over HRAS^G12V⁺ cells, seemingly scanning their surface (Fig. 2j). Cell footprints of those neutrophil movements trace the outline of HRAS^G12V⁺ clusters (Fig. 2i). Notably,

**Fig. 1 | mmp9:Citrine identifies mature neutrophils. a** Confocal images of triple transgenic larvae *Tg(lysC:CFP-NTR)[vi002]/ Tg(BACmmp9:Citrine-CAAX)[vi003]/ Tg(mpeg:mCherry)[gl23]* reveal myeloid cells with different expression levels of the three analyzed markers. Stitched whole-mount images of a 5 dpf (days post fertilization) larva. *n* = 3. **b** High magnification showing presence of three different cell types: #1 neutrophil: *lysC:CFP[+]/ mpeg:mCherry[+]/ mmp9:Citrine[+]*; #2 neutrophil: *lysC:CFP[+]/ mpeg:mCherry / mmp9:Citrine[-]*; #3 macrophage: *lysC:CFP / mpeg:mCherry[+]/ mmp9:Citrine[-]*. **c** Flow cytometric analysis (BD LSRFortessa) of myeloid cells from *mpeg:mCherry / lysC:CFP[+]* control larvae, *lysC:CFP / mpeg:mCherry[+]* controls, or triple transgenic larvae *lysC:CFP / mmp9:Citrine[+]/ mpeg:mCherry[+]* (dot plots left to right) at 3 dpf. Histogram and box plot showing expression of *mmp9:Citrine* in myeloid cells (combined gates of Q1, Q2, Q3; Minimum = 12.4%, Maximum = 17.1%, Median = 15.7%; Box 25th–75th percentile). Violin plots indicating the distribution and expression levels of *mmp9:Citrine* among myeloid populations (*mmp9:Citrine[+]* gate; *n* = 5 with pools of 15 triple transgenic larvae each). mfi = mean fluorescence intensity (**d**) *mmp9:Citrine[+]* neutrophils (red histogram) have a higher side scatter (SSC) and therefore granularity compared to *mmp9:Citrine[-]* neutrophils (blue histogram). Flow cytometric analysis of cells isolated from *Tg(lysC:CFP-NTR)[vi002]/ Tg(BACmmp9:Citrine-CAAX)[vi003]* at 2 dpf. Cells were gated on the *lysC:CFP[+]/mmp9:Citrine[+]* or *lysC:CFP[+]/mmp9:Citrine[-]* populations and analyzed for SSC (*n* = 3). **e** Cells were isolated from kidneys, spleen or blood of adult *Tg(lysC:dsRed)[nz50Tg]/ Tg(BACmmp9:Citrine-CAAX)[vi003]* and analyzed by flow cytometry for frequency of Mmp9[+] cells. **f** Neutrophils were isolated from whole kidney marrow (WKM) of adult *Tg(lysC:dsRed)[nz50Tg]/ Tg(BACmmp9:Citrine-CAAX)[vi003]* zebrafish and sorted into *lysC:dsRed[+]/ mmp9:Citrine[+]* or *lysC:dsRed[+]/ mmp9:Citrine[-]* populations, cytospins were prepared and Pappenheim stained. Different neutrophil maturation stages were scored blinded, showing that the *lysC:dsRed[+]/ mmp9:Citrine[+]* population consists of highly differentiated neutrophil stages. INT = intermediate; HI = high. Graph presents average percentages of *n* = 4 kidneys (721 Mmp9[+] cells and 819 Mmp9[-] cells were analyzed in total). **g** Representative ultrastructural images show an unsegmented nucleus and round and elongated granules in *lysC:CFP[+]/ mmp9:Citrine[-]* cells (*n* = 5 from one TEM transmission electron microscopy experiment) contrasting with a segmented nucleus and predominantly elongated granules in *lysC:CFP[+]/ mmp9:Citrine[HI]* cells (*n* = 2); *, round granules; #, elongated granules.

this spreading behavior around HRAS[G12V+] cells was significantly enriched in Mmp9[HI] cells (mean 56.5%; *n* = 8; one-way ANOVA, *P* = 0.0022) compared to Mmp9[INT] (17.3%) or Mmp9[-] (17.9%) cells (Fig. 2k, Supplementary Movie 3).

In summary, we observed that *mmp9* transgene expression correlates with a high maturation grade in neutrophils with increased effector functions, such as recruitment and phagocytosis during bacterial infections and augmented interactions with pre-neoplastic cells.

## Single-cell transcriptomics defines zebrafish neutrophil maturation stages

To scrutinize neutrophil maturation in zebrafish at the molecular level and to enable cross-species comparison to mammals, we performed droplet-based scRNA-seq on WKM and on LysC[+] neutrophil populations expressing different Mmp9 levels (NO, INT, HI) sorted from the WKM of two adult *Tg(lysC:CFPNTR)[vi004]/ Tg(BACmmp9:Citrine-CAAX)[vi003]* zebrafish (Fig. 3a, Supplementary Fig. 3a). The generated dataset comprised a total of 18,150 cells passing quality control (Fig. 3b, Supplementary Fig. 3b–g, Supplementary Data 9).

Sorted neutrophil populations and neutrophils from WKM overlapped in our dataset, indicating that our sorting strategy captured all neutrophils present in WKM (Fig. 3b). To investigate the maturation dynamics, we focused on cells consistently annotated as neutrophils (*n* = 15,876 cells) based on a bioinformatic mapping to two independent reference datasets[24,38] (Fig. 3c, d) and used the Slingshot algorithm to infer the structure of the underlying trajectory (Fig. 3e, f)[39]. Consistent results were also obtained with other trajectory inference algorithms (Supplementary Fig. 4) and the validity and directionality of this trajectory was supported by (i) the sequential order of the sorted subpopulations along the trajectory (in order: Mmp9[NO], Mmp9[INT], Mmp9[HI]; Fig. 3f), (ii) decreasing levels of lysozyme C (*lyz)* and increasing levels of *mmp9* (Fig. 3g), and (iii) a decrease in the number of cycling cells (inferred bioinformatically using the CellCycleScoring function from the Seurat package; Supplementary Fig. 3h). A cluster separated from the main neutrophil population was mainly formed by cells in the G2/M phase (labeled by *, Fig. 3b, Supplementary Fig. 3e, k, l, m).

We used the tradeSeq[40] algorithm to define the genes associated with neutrophil maturation ($P_{adj}$ < 0.05, top 1500 in descending order based on Wald-statistic; Supplementary Data 1). Based on expression patterns of these genes, we partitioned the cells into four maturation phases along the trajectory (P1 = early, P2 = early/intermediate, P3 = intermediate/late, P4 = late). Similarly, we partitioned the genes into three modules (M1 = early, M2 = intermediate, M3 = late) (Fig. 4a). While M1 and M2 were almost exclusively expressed in phase P1 and P2, respectively, M3 genes were expressed in both P3 and P4 (Fig. 4b).

Functional enrichment analysis using hypeR[41] indicated that earlier modules M1 and M2 related to cell cycle and proliferation while module M3 comprised genes associated with functions of mature neutrophils like migration, immune activation, and inflammation (Fig. 4c, Supplementary Data 3).

We next examined genes associated with certain neutrophil functions in detail (Fig. 4d)[6,8,12,14]. First, we found proliferation factor *myca* expressed during early maturation (P1), while advancing maturation (P3, P4) was associated with anti-apoptotic genes such as *mcl1a*, *mcl1b* and *mxi1*, a negative regulator of *myc*[42]. Second, we detected genes related to oxidative stress response (*selenoh*[43], *abcc13*, *prdx1*) early in P1, which could aid to protect progenitor cells from oxidative damage. Third, we found a down-regulation of the marrow retention factor *cxcr4b* in P4, and conversely an upregulation of *cxcr1*, *lyn*, and the migration-related transcription factor *atf3* in P4, in line with a putative switch towards a migratory phenotype[44].

Finally, we also found genes encoding granule-related proteins expressed in specific phases (Fig. 4d), these granules have conventionally been used for neutrophil staging in mammals[5]. At the beginning of the trajectory *lyz* from P1 on (compare Fig. 3g), primary granule-related genes *cd63*, *srgn*, and *mpx* from P2 on, secondary granule-related NADPH oxidase subunits *cybb* and *cyba* from P3 on, and the gelatinase *mmp9* and *mmp13a.1* (a putative orthologue of human *MMP1*) in P4. The expression of these genes is consistent with a staging of human neutrophils from myeloblasts (P1) through promyelocytes (P2) and myelocytes/metamyelocytes (P3) to banded/segmented neutrophils (P4). The last phase (P4) was also associated with the upregulation of the pro-inflammatory cytokine *il1b* (Fig. 4d), confirming data from mouse neutrophils[12,14].

Taken together, our data demonstrate that neutrophils in the kidney marrow of zebrafish mature in a continuous process advancing from a proliferative stage to a post-mitotic, anti-apoptotic, and migratory phase. Detection of genes associated with certain granule types suggests a similar sequence of granule production as in mammals.

## C/ebp-β governs expression of late granule genes during maturation

Next, we sought to identify the key regulators of the neutrophil maturation process in zebrafish. Many known neutrophil TFs displayed dynamic expression patterns along the trajectory (Fig. 5a)[8]. We reasoned that the expression of key regulatory TFs was closely correlated with the expression of their target genes, albeit possibly with a temporal delay. We therefore compared TF expression with the expression of genes in each module using dynamic time warping (DTW) analysis, which highlighted *ybx1*, also known as key splicing factor in

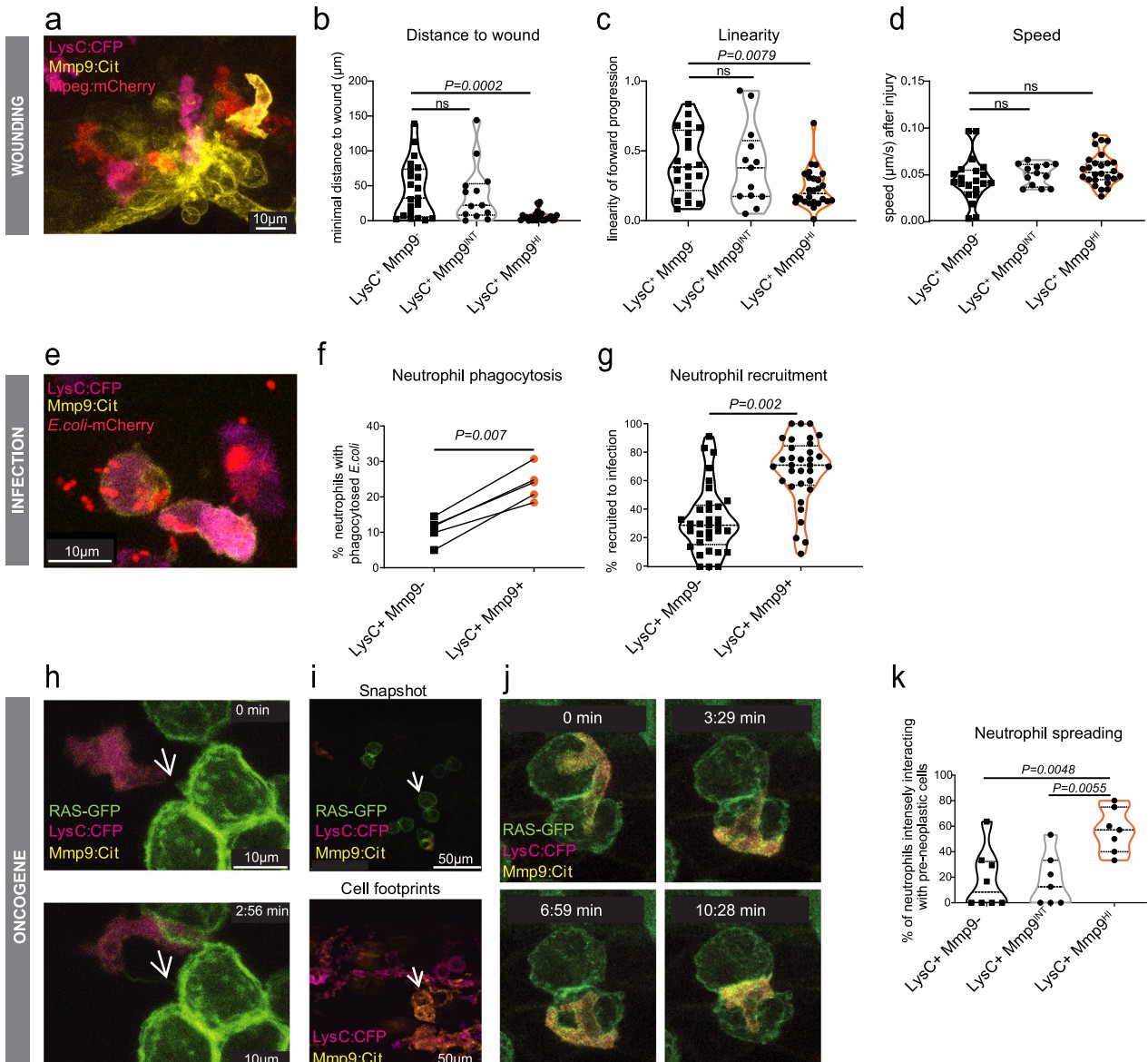

**Fig. 2 | Mmp9⁺ neutrophils show functions of mature neutrophils. a–d**
Recruitment of neutrophils to an injury of the fin at 2 dpf (days post fertilization) *lysC:CFP⁺ mmp9:Citrine^HI*, *lysC:CFP⁺ mmp9:Citrine^INT* and *lysC:CFP⁺ mmp9:Citrine⁻* neutrophil tracks (*n* = 21; 13; 26 cells, df = 57, respectively; *n* = 6 larvae; One-way ANOVA with Dunnett's test) were analyzed over a 2 h period: **b** for distance to a wound ROI (*P* = 0.0002, *F* = 9.955), (**c**) linearity of forward progression (mean straight line speed/ track mean speed) (*P* = 0.0086, *F* = 5.172) and (**d**) speed (*P* = 0.1315, *F* = 2.103). ns = not significant. INT = intermediate expression; HI = high expression. **e−g** In vivo phagocytosis assays were performed by injecting mCherry-labeled *E. coli* into the caudal vein or otic vesicle of *Tg(lysC:CFP-NTR)^yi002/ Tg(BACmmp9:Citrine-CAAX)^yi003* zebrafish larvae at 2 dpf. **e** *E. coli* were observed inside both, *lysC:CFP⁺ mmp9:Citrine⁺ and lysC:CFP⁺ mmp9:Citrine⁻* neutrophils 6 hpi (hours post infection). **f** Phagocytosis experiments analyzed by flow cytometry (*n* = 5, each approx. 20 larvae; two-tailed paired *t* test, *P* = 0.007; df = 4); **g** Neutrophil recruitment to *E.coli*-mCherry injected into the otic vesicle of 3 dpf larvae was analyzed by confocal microscopy (*n* = 33, two-tailed paired *t* test, *P* = 0.002, df = 32). **h–k** Neutrophil- pre-neoplastic cell interactions were observed by confocal microscopy in *Et(kita:GAL4)^hzm1/ Tg(UAS:EGFP-HRAS_G12V)^io006/ Tg(lysC:CFP-NTR)^yi002/ Tg(BACmmp9:Citrine-CAAX)^yi003* zebrafish larvae starting at 78 hpf. **h** Still Images of z-stack maximum projections from a time-lapse movie showing Mmp9⁺ neutrophils forming dynamic contacts with GFP⁺ kita tumor cells. White arrows point at a GFP⁺ cell tether. Z-stacks were acquired every 88 s. **i** Snapshots and cell footprints were taken from the same time-lapse movie (*t* = 5 h). Superimposition of *lysC:CFP* and *mmp9:Citrine* from all time frames generating neutrophil footprints (bottom). White arrow points out how the movements of an Mmp9⁺ neutrophil copied the outline of the RAS-GFP⁺ cluster seen in the snapshot (top). **j** Close-up clippings showing an Mmp9⁺ neutrophil spreading over the pre-neoplastic cell cluster marked by a white arrow in (**i**). **k** Quantification of neutrophil-pre-neoplastic interactions were performed from maximum projections of different time-lapse movies. The frequency of interacting neutrophils of each sub-population (no, intermediate or high *mmp9:Citrine* levels) getting into close, intense interactions with GFP⁺ kita/RAS skin pre-neoplastic cells. (*n* = 8, one-way ANOVA, *P* = 0.0022, *F* = 8.561, df = 19).

hematopoietic development[45], *hmgb2b* and *dnajc1* as top regulators of early maturation genes in module M1 (Fig. 5b, c; Supplementary Fig. 5; Supplementary Data 4, 5). In contrast, *cebpb* and *atf3* emerged as the top-ranked transcription factors for late maturation (M3). C/EBP-β governs demand-driven granulopoiesis during infection in zebrafish and in mammals[46,47] and has also been identified as a late-expressed transcription factor in neutrophils in mice[12,14]. However, a role during steady-state neutrophil maturation has not been described to date.

We hypothesized that C/ebp-β is involved in the regulation of neutrophil maturation in zebrafish. After confirming that *cebpb* expression in sorted LysC⁺Mmp9⁺ neutrophils was significantly higher than in LysC⁺Mmp9⁻ neutrophils and unsorted cells (repeated

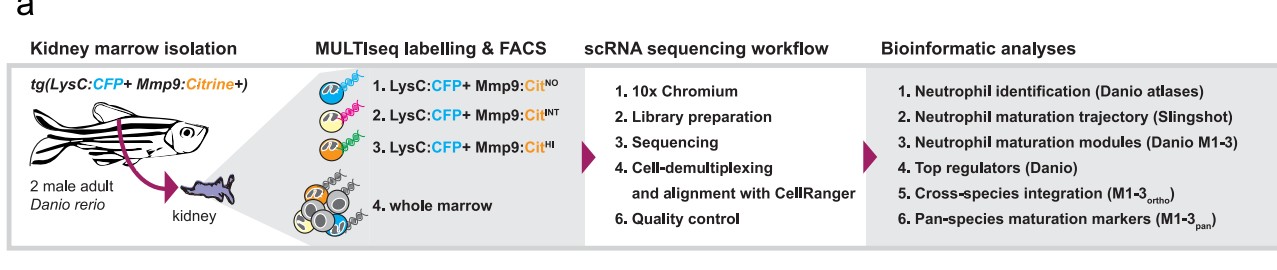

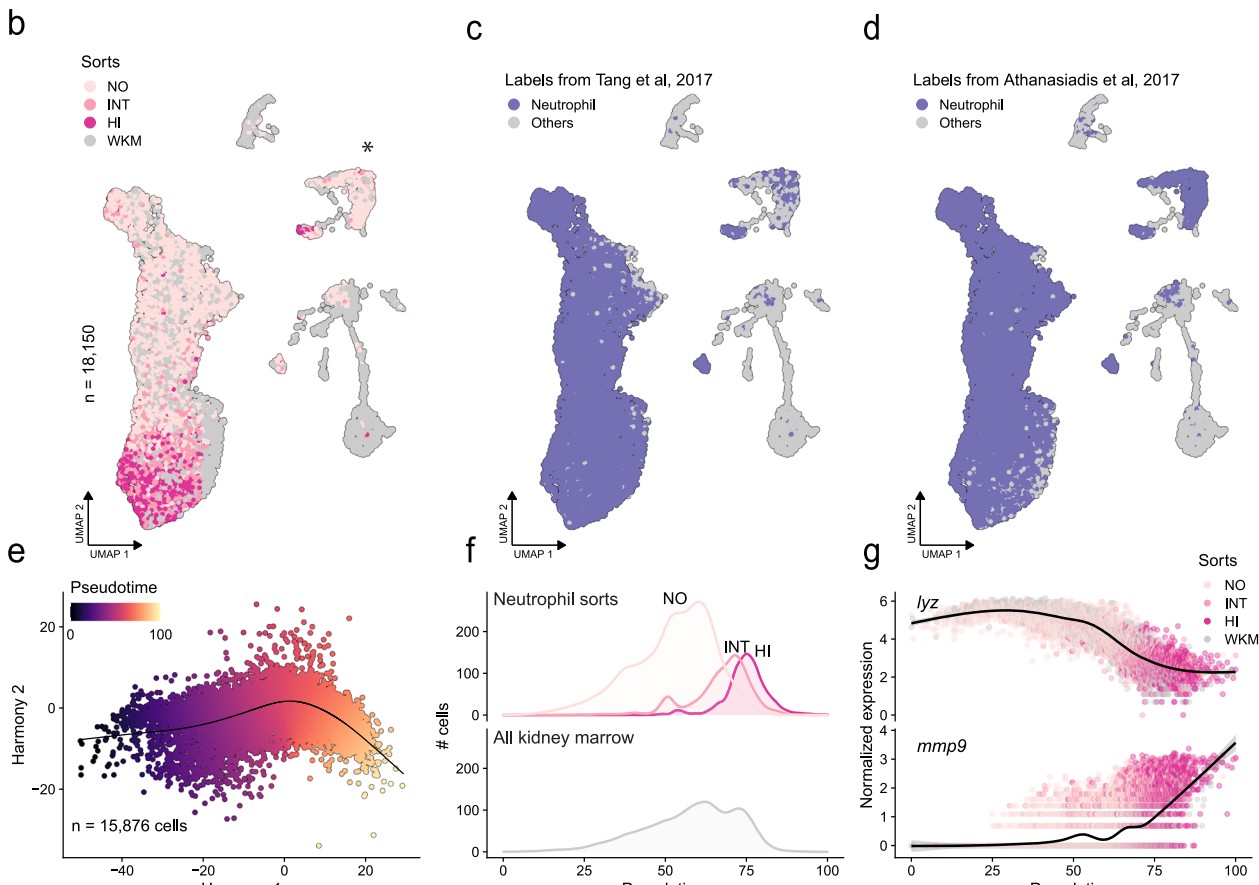

**Fig. 3 | scRNA-seq of zebrafish neutrophils reveals a continuous maturation process. a** Schema showing the workflow for cell isolation, multiplexing and scRNA-seq, as well as the bioinformatics analyses applied. NO = no expression; INT = intermediate expression; HI = high expression. WKM = whole kidney marrow. **b** Uniform Manifold Approximation and Projection (UMAP) of single-cell RNA-seq data (*n* = 18,150 cells) showing cells from FACS-sorted neutrophil populations (lysC⁺/ mmp9 = NO, INT, HI) and unsorted whole-kidney marrow cells. The cluster labeled with an asterisk (*) represents a group of cells dominated by cell cycle effects (cp. Supplementary Fig. 3k–m). **c**, **d** Reference-based labeling of cell types

by projecting cells to two zebrafish hematopoietic reference atlases in the same UMAP as in panel **b**[24,38]. As expected, a strong overlap with genetically labeled *Tg(mpx:GFP)* neutrophils is observed (highlighted in color). **e** Harmony[82] plot of the inferred neutrophil maturation trajectory in zebrafish (*n* = 15,876 cells). A continuous trajectory indicating a maturation continuum is observed. **f** Line plots of the distribution of sorted (NO, INT, HI) and unsorted neutrophil populations along the inferred trajectory. **g** Smoothed expression of *lyz* and *mmp9* in neutrophils along the inferred trajectory.

measures ANOVA, *P* < 0.05, Fig. 5d), we targeted *cebpb* translation using a published AUG-binding morpholino (Fig. 6a). No systemic adverse effects were observed, knockdown did not influence steady-state neutrophil numbers, in line with previous data (Fig. 6b)[46]. We examined whether C/ebp-β regulates the augmented phagocytic function of Mmp9⁺ neutrophils but found no effect on uptake of mCherry-labeled *E.coli* (Fig. 6c). *Cebpb* knock down affected expression of the published C/ebp-β target genes[14], *mmp9* and *fcer1gl*, but not of *spi1b* and *mcl1a* in FACS-sorted LysC⁺Mmp9⁺ neutrophils (Fig. 6d). Strikingly, we found that *cebpb* morpholino treatment reduced the frequency of LysC⁺Mmp9⁺ neutrophils at 2 and 3dpf (Fig. 6e), indicating an instructive role of C/ebp-β in the differentiation towards mature Mmp9-expressing neutrophils. Conversely, *cebpb* RNA

overexpression led to higher LysC⁺Mmp9⁺ neutrophil frequencies and mmp9 expression levels, further supporting this role (Fig. 6f). To confirm the function of C/ebp-β in adult neutrophils we generated *cebpb*^MUT vi006^ fish by CRISPR-Cas9 introducing a 76 bp deletion plus a 2 bp insertion leading to a frameshift and premature stop codon (Supplementary Fig. 6a, b). In agreement with the results in larvae, we observed a strongly diminished LysC⁺Mmp9⁺ population in the WKM of homozygous *cebpb*^MUT vi006^ fish (in *mmp9:Cit*^HOM^ mean = 1.4%; in *mmp9:Cit*^HET^ mean = 5.1%) compared to *cebpb*^WT^ (in *mmp9:Cit*^HOM^ mean = 32%; in *mmp9:Cit*^HET^ mean = 25.1%) (Fig. 6h), but no change in the frequency of LysC⁺ neutrophils (Fig. 6i). Together, these data indicate that C/ebp-β – in addition to its role in demand-driven hematopoiesis – regulates aspects of steady-state neutrophil

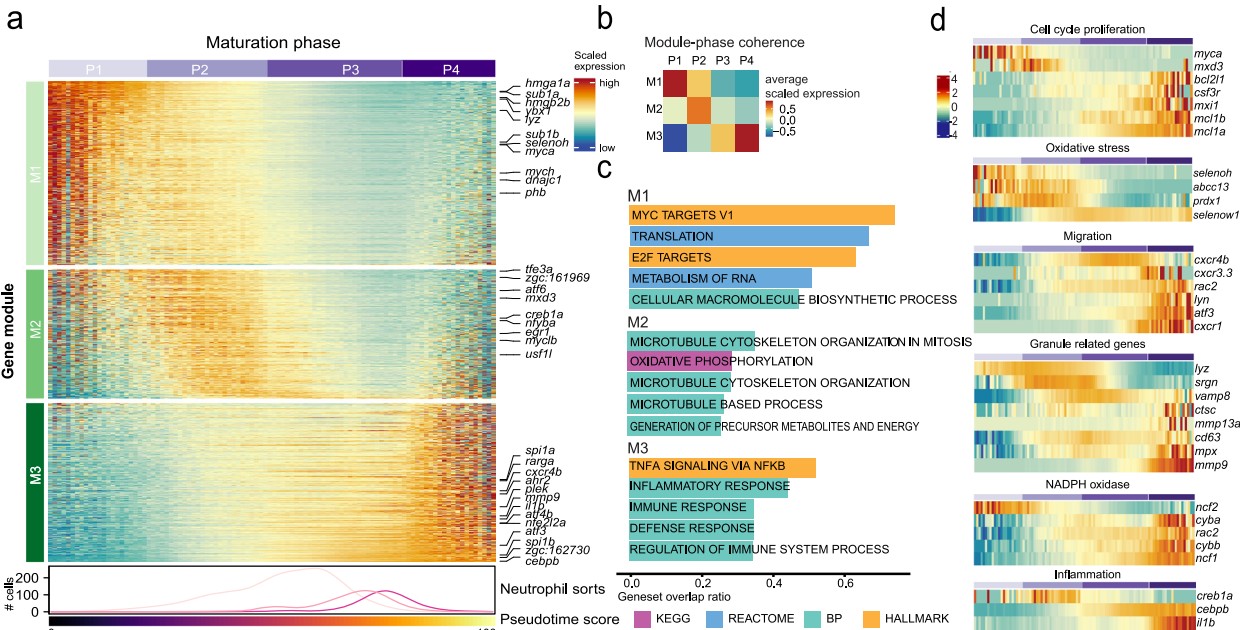

**Fig. 4 | Trajectory analysis uncovers the underlying cell phases and governing gene modules. a** Heatmap showing differentially expressed genes (tradeSeq;[40] $P_{adj} < 0.001$, top 1500 based on Wald-statistic) along the maturation trajectory. Cells have been allocated to four distinct phases (P1-P4) and into three distinct gene modules (M1-M3) by hierarchical clustering. The columns of the heatmaps correspond to 100 ordered discrete bins covering the maturation trajectory. The bottom line-annotation shows the distribution of sorted subpopulations along the maturation trajectory (see Fig. 3f) and the bar shows the pseudotime score (see Fig. 3e). **b** Heatmap summarizing average gene expression per module and phase (from **a**). **c** Top-5 enriched gene sets per module from the indicated source pathway databases (hypeR[41] over-representation analysis). **d** Heatmaps of differentially expressed genes (excerpts from **a**) associated with selected neutrophil-related functional pathways.

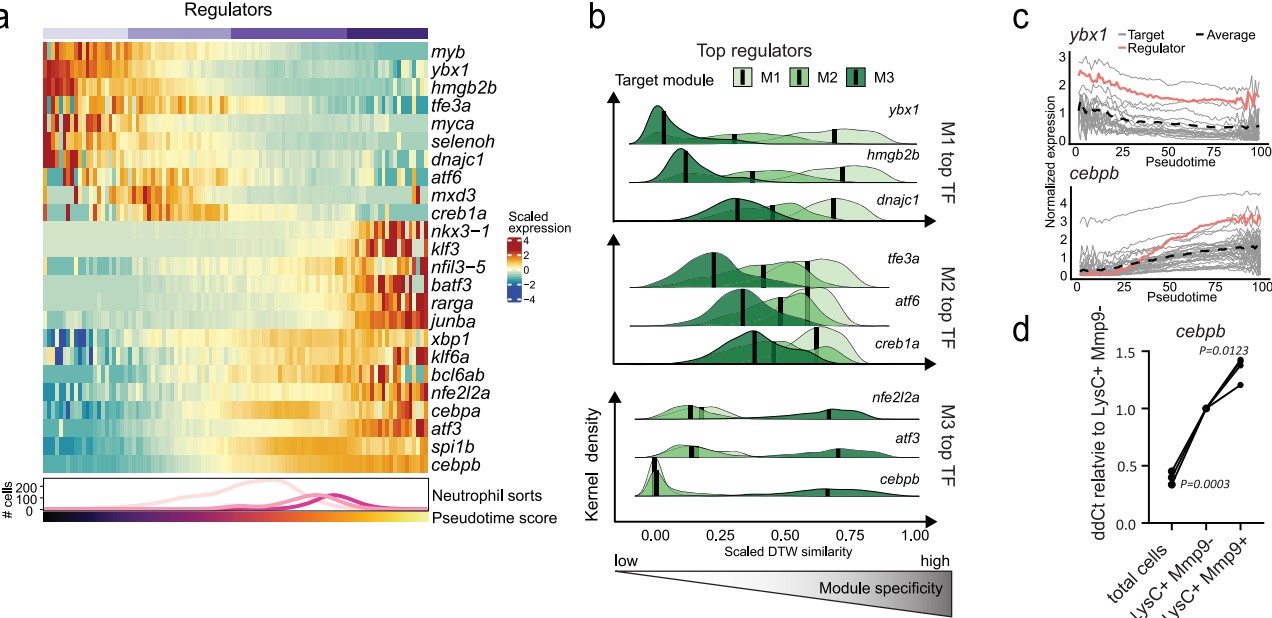

**Fig. 5 | Cebpb expression is associated with neutrophil maturation in zebrafish. a** Heatmap showing regulation of selected transcription factors (TFs) during neutrophil maturation. The bottom line-annotation shows the distribution of sorted subpopulations along the maturation trajectory (see Fig. 3f) and the bar shows the pseudotime score (see Fig. 3e). **b** Ridge plot showing top candidate regulators and their specificity to target genes in each module. Here, "module specificity" measures the similarity of the expression of a TF to all target genes in the neutrophil maturation modules compared to other TFs. It was calculated as 1 minus the dynamic time warping (DTW) distance and scaled relative to all other TFs. Thus, a high score on the x-axis indicates a target gene with a closely matching and specific expression pattern for the TF. The y-axis shows the distribution density for all genes in each module (M1, M2, M3; different colors). **c** Expression of top TFs *ybx1*, compared to its putative target genes in module M1 (top), and *cebpb*, compared to genes in M3 (bottom). **d** qPCR validation of *cebpb* expression in sorted *lysC:CFP⁺/ mmp9:Citrine⁺ and lysC:CFP⁺/ mmp9:Citrine⁻* neutrophils. (*n* = 3, each sample sorted from a pool of 160 larva; repeated measures ANOVA).

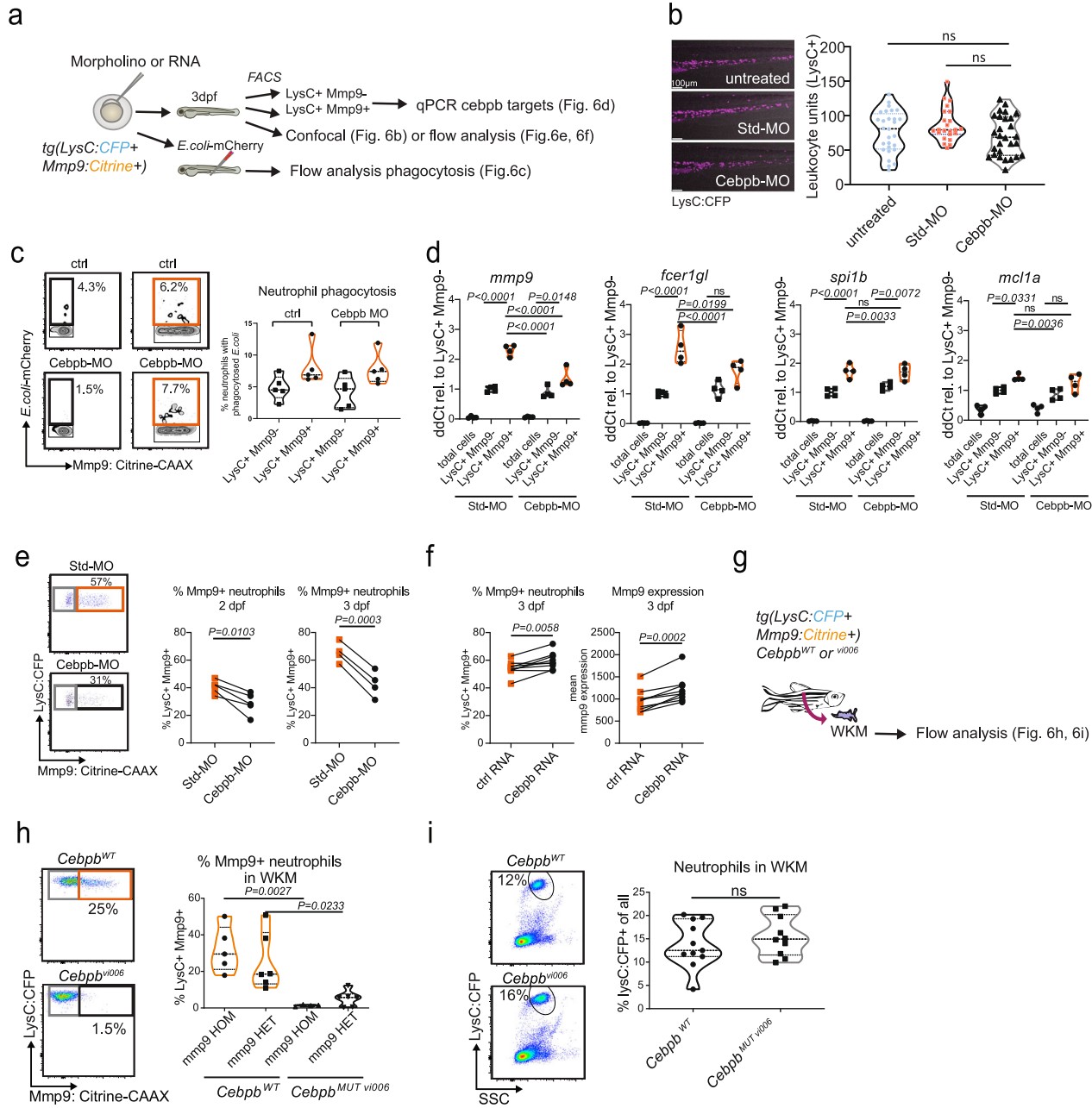

**Fig. 6 | Cebpb regulates aspects of late neutrophil maturation in zebrafish.**
**a**–**e** Analyses of standard (Std)-morpholino (MO) or *cebpb*-morpholino treated *Tg(lysC:CFP-NTR)^vi002/ Tg(BACmmp9:Citrine-CAAX)^vi003* larvae at 3 dpf (days post-fertilization). **a** Schematic drawing of subsequent experiments. **b** ImageJ analysis of LysC⁺ leukocyte numbers in the tail region of untreated, control (ctrl) standard morpholino or *cebpb* morpholino-injected larvae (n = 28; 23; 28 larvae, respectively, one-way ANOVA with Tukey´s test, P = 0.1343, F = 2.041, representative image shown to the left). Leica SP8 confocal images with a HC PL APO CS 10x/0.40 DRY; Zoom 0.85x. **c** In vivo phagocytosis assay with *E.coli*-mCherry injected into the caudal vein after *cebpb* morpholino treatment. Representative flow cytometry plots (left). **d** qPCR of selected target genes of C/ebp-β (selected from Xie et al.[14].) in FACS-sorted cells (n = 4; sorted from 65-102 larvae each; one-way ANOVA with Tukey´s test; all P < 0.0001, df = 18) after morpholino treatment at 3 dpf. ns = not significant. **e** Representative flow cytometry plots show reduction in LysC⁺Mmp9⁺ neutrophils after *cebpb* morpholino treatment (left). Plots summarizing four

independent experiments each with pools of approx. 20 larvae per group analyzed by flow cytometry at 2 and 3 dpf after morpholino treatment (right). Two-tailed paired *t* test. **f** Frequencies of LysC⁺Mmp9⁺ neutrophils and mean *mmp9:Citrine* expression are increased after *cebpb* full-length RNA overexpression. Analyzed by flow cytometry 3 dpf after mRNA injection (n = 7; pools of 20 larvae each), two-tailed paired *t* test. **g**–**i** Analyses of neutrophils from adult Cebpb mutant fish. WKM = whole kidney marrow. **g** Schematic drawing. WKM of adult *Cebpb* wildtype (*cebpb^WT*) or *Cebpb* mutant (*cebpb^MUT vi006*) *Tg(lysC:CFP-NTR)^vi002/ Tg(BACmmp9:Citrine-CAAX)^vi003* fish was evaluated by flow cytometry. **h** Representative dot plots (left). Violin plot summarizing percentages of LysC⁺Mmp9⁺ neutrophils of all LysC⁺ neutrophils in *mmp9:Cit^HOM* (n = 5) *mmp9:Cit^HET* (n = 6) *cebpb^WT* and *mmp9:Cit^HOM* (n = 4) *mmp9:Cit^HET* (n = 6) *cebpb^MUTvi006* kidneys. (One-way ANOVA with Tukey´s test P = 0.0006, F = 9.656, df = 17). **i** Representative dot plots (left). Violin plot summarizing percentages of LysC⁺ neutrophils of all live cells in *cebpb^WT* (n = 11) and *cebpb^MUT vi006* (n = 10) kidneys (two-tailed unpaired *t* test; P = 0.3797).

maturation such as generation of mature Mmp9+ neutrophils and expression of granule proteins but not their phagocytic function.

### Cross-species comparison allows staging of zebrafish neutrophil maturation phases and identifying conserved gene signatures

Next, we wanted to gauge the conservation of neutrophil maturation trajectories between zebrafish and mammalian species, an important piece of information for investigating and modeling granulopoiesis across species. To address this point, we collected transcriptome datasets capturing progenitor, early, and late neutrophil maturation stages in human and mice. We included both in vivo and in vitro data in this selection[11,13,48–51]. We then followed two complementary approaches to assess similarities at the cellular and molecular level.

First, we integrated all datasets with the four phases of neutrophil maturation that we defined in zebrafish (aggregating transcriptome profiles of single cells per phase) and used unsupervised hierarchical clustering to group similar stages in an unbiased manner (Fig. 7a, Supplementary Data 10; zebrafish phases P1-P4 highlighted with dotted lines). We found that phase P1 clustered together with early stages of human in vitro differentiation (Neu3h-12h[51]) and unipotent neutrophil precursors and early neutrophils in mice (ly6c_neg_gmp[13], preNeu[11], proNeu-1/2[50], proneu1[13]) and human (NCP1-4[48], eNeP[49]). Phase P2 corresponds to the transition stage that includes preNeu-1[50] and proneu2[13] in mouse and promyelocytes (PM)[48], myelocytes (MY)[48] and N1-neutrophils excluding the early progenitor portion ("N1 w/o eNeP")[49] in human. Finally, both P3 and P4 phases clustered with cells reaching the end of in vitro differentiation (Neu96h)[51] and the differentiated and *mmp9* positive mouse neutrophils described by Muench et al. 2020 (immNeu)[50], as well as banded cells (BC)[48]. Other cells captured from peripheral blood (blood_neu[11], mature_neu[11], Neuts[49]) as well as segmented neutrophils (SN)[48] and PMN[48] also clustered with the zebrafish neutrophils at stages P3-P4 although at higher distance, which is consistent with ongoing maturation after exiting the bone marrow, a process missing in the zebrafish WKM neutrophil data. Altogether, this analysis indicated that the four zebrafish phases mirrored major stages of neutrophil maturation in mouse and human.

Second, we asked whether the neutrophil maturation followed similar dynamics in all species at the transcriptomic level by comparing the maturation trajectory from our zebrafish model (Fig. 7b) to those from mouse[10,12,14] (Fig. 7c) and human[52–54] (Fig. 7d) based on published bone marrow scRNA-seq data (Supplementary Data 10). For each trajectory, we aligned the expression patterns of genes along the maturation trajectory to the corresponding homolog along the zebrafish trajectory using cross-correlation (Fig. 7e, f). We found that gene expression in early development (M1$_{ortho}$) was highly congruent with their mouse and human orthologues in all datasets. At later stages (M2$_{ortho}$ and M3$_{ortho}$ modules), expression profiles diverged, but a subset of genes in each module was found conserved across all datasets (Fig. 7e, f). For instance, this analysis confirmed the strong agreement of identified candidate regulators of M1 (*ybx1*) and M3 (*cebpb*) across the three species (Supplementary Fig. 7a). *mmp9* expression lagged slightly behind in zebrafish compared to some human (Tabula Sapiens and Xie 2021) and mouse (Grieshaber–Bouyer 2020) datasets that contain peripheral neutrophils that were not analyzed in our WKM isolates. Conversely, other genes showed species-specific differences; for instance, *tgfbi* was expressed earlier in human but later in mouse trajectories compared to zebrafish. Other genes showed a lag (e.g., *txnipa*) or advance (e.g., *amd1*) compared to zebrafish. To assess the similarity of maturation trajectories at species-level, we calculated the mean cross-correlation coefficient per dataset for each module (Fig. 7e, f, Supplementary Fig. 7b) and used hierarchical clustering to group datasets with similar maturation dynamics (Supplementary Fig. 7c–e). We found that the range of cross-correlation lags for M1$_{ortho}$ and M2$_{ortho}$ was smaller than for M3$_{ortho}$, for which differences between all datasets increased. Taken together,

these results indicate a strong conservation (across all species) of early neutrophil maturation, while differences arise at late maturation stages.

As an external validation, we sought to compare our orthologous gene modules (M1$_{ortho}$-M3$_{ortho}$) to frequently used gene signatures for human immature neutrophils (HAY_BONE_MARROW_IMMATURE_-NEUTROPHIL; M39200)[55]. Surprisingly, the majority of putative "immature neutrophil" genes were attributable to late modules (M2$_{ortho}$, M3$_{ortho}$) of zebrafish neutrophil maturation (Supplementary Fig. 8a), but also to late time points of human in vitro neutrophil differentiation (Supplementary Fig. 8b), highlighting the necessity for alternative gene signatures of different maturation grades. Our cross-species comparison allowed us to define a pan-species gene signature of neutrophil maturation (absolute cross-correlation lag <= 50; M1$_{pan}$ = 304, M2$_{pan}$ = 176, M3$_{pan}$ = 122; Supplementary Data 1, 6; see *Cross-species integration and comparison* for details).

Finally, we went on to show the utility of the pan-species neutrophil maturation signature by using it to infer the maturation stage of neutrophils in heterogeneous tissues from bulk RNA sequencing data. To this end, we analyzed metastatic neuroblastoma samples[56]. Neuroblastoma is a childhood cancer derived from the sympathetic nervous system that frequently disseminates into the BM. Comparison of BM with ($n = 17$ datasets) and without tumor cell infiltration ($n = 21$ controls) thus presents an in vivo test case to examine our gene modules. Single-sample Gene Set Enrichment Analysis (ssGSEA)[57] indicated a differential enrichment of early modules (M1$_{pan}$ and M2$_{pan}$) in control samples opposed to an enrichment of the mature module M3$_{pan}$ in tumor-infiltrated samples (Fig. 8a, Supplementary Fig. 9a). For validation we scored the percentage of segmented neutrophils on BM cytospins from patients with localized ($n = 9$) and infiltrated ($n = 12$) neuroblastoma by imaging mass cytometry (IMC; Iridium-intercalator = nuclear; CD15 = granulocytic; Fig. 8b, c, Supplementary Fig. 9b). This confirmed a significant increase of segmented neutrophils in the metastatic group compared to control (Unpaired $t$ test; $P = 0.025$).

In summary, we established a comprehensive cross-species transcriptome comparison of neutrophil maturation, suggesting a high degree of conservation between zebrafish and mammalian models. The pan-species gene signature derived from this comparison will present a valuable and robust alternative to existing gene signatures of neutrophil maturation and is applicable for examination of human samples.

## Discussion

In this study, we generated *lysC:CFP/mmp9:Citrine* transgenic zebrafish, which enabled us to identify and study neutrophils of different maturation grades non-invasively in a living organism. The importance of neutrophils is underlined by their long evolutionary conservation: Granular cells are present already in invertebrates such as lancelets (*Branchiostoma*), and myeloperoxidase is even produced by invertebrates[58]. However, to use animal models for human disease, it is crucial to first delineate similarities and differences between species. Our transgenic zebrafish made it possible to separate neutrophils into populations that differed in maturation-dependent neutrophil functions (e.g., phagocytic capability, recruitment towards bacterial infections, and the ability to interact with oncogene-expressing cells). As antibody markers for maturation states are not available in zebrafish, our Mmp9-reporter was thus instrumental in dissecting these different cell populations. Notably, among myeloid cells, Mmp9 expression was almost completely restricted to neutrophils (LysC+ Mpeg+ and LysC+ Mpeg-) in our model and only found in a small population of steady-state macrophages (LysC- Mpeg+) at low levels. In interaction with malignant cells, macrophages might start expressing Mmp9 as observed[59,60].

Powered by the new model, our subsequent scRNA-seq analyses revealed a continuous neutrophil maturation trajectory in zebrafish

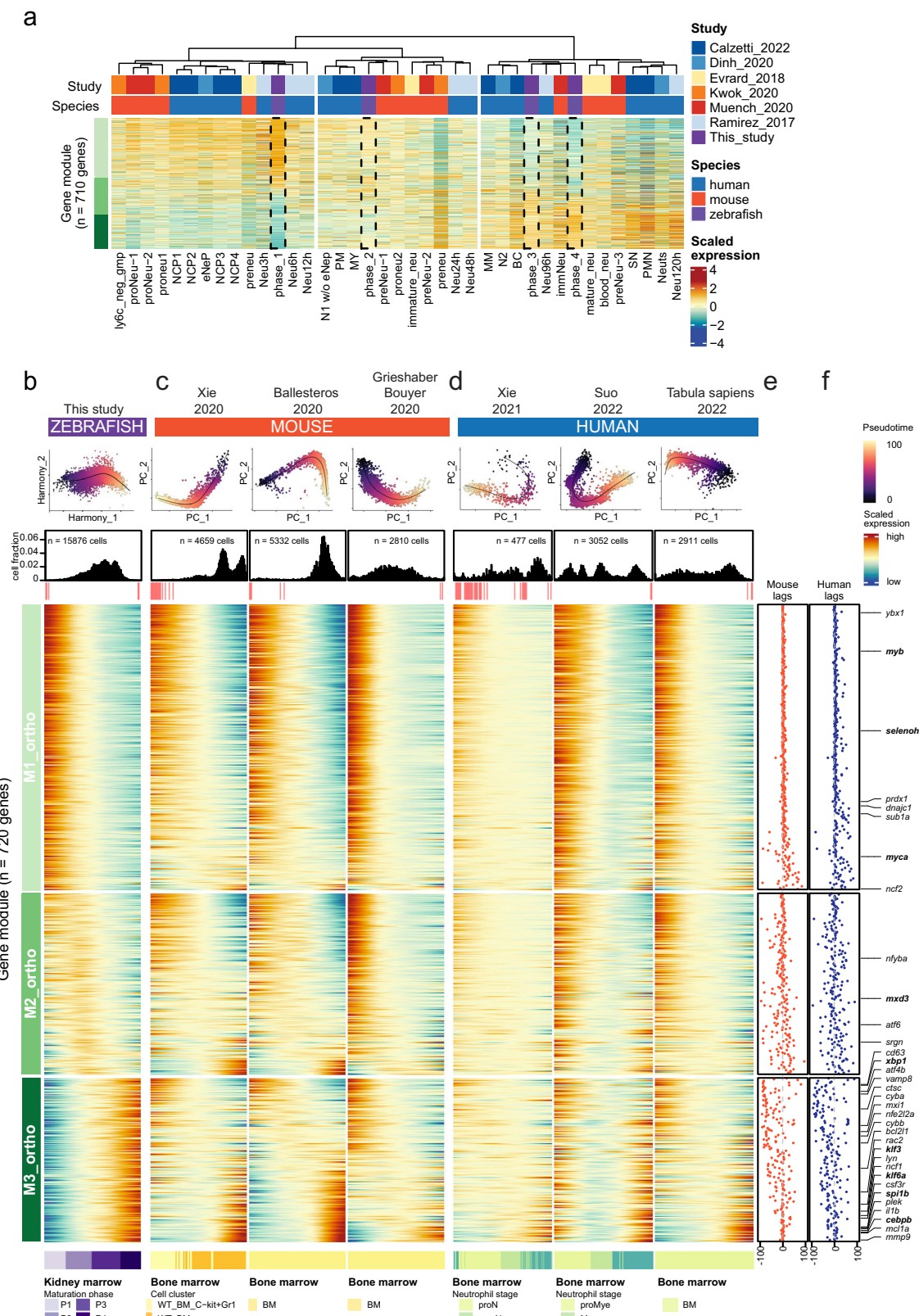

akin to what has been observed in mice[12]. We found that maturing zebrafish neutrophils went through multiple stages equivalent to those in mammals (Fig. 9), including also cells mimicking early, unipotent progenitors recently described in mice[13,50] and human[48,49]. All three species opted for similar gene expression programs at these early stages, while transcriptomic differences became more pronounced

with increasing maturity. For instance, the expression of transcription factors *ybx1* and *cebpb* were particularly well synchronized across species, while the expression of *tgfbi* diverged. As in mice we found late maturation under the control of the transcription factor *cebpb* and associated with inflammatory pathways and increasing levels of *il1b* and *csf3r* suggesting a pro-inflammatory state of neutrophils awaiting

**Fig. 7 | Alignment of expression trajectories across zebrafish, mouse and human reveals concordant and divergent stages of maturation. a** Heatmap of hierarchically clustered neutrophil maturation stages across studies[11,13,48–51] based on zebrafish neutrophil maturation signatures (M1ortho, M2ortho, M3ortho). The four aggregated zebrafish phases are highlighted by a dotted box. Column annotations on the top show dataset annotation and row annotations on the left show module membership of each gene. Panels display zebrafish kidney marrow (**b**), mouse (**c**) and human (**d**) bone marrow (BM) scRNA-seq data ordered by their own maturation trajectory[10,12,14,52–54]. Ordered from top to bottom, column annotations display dataset annotations, inferred trajectories per dataset with pseudotime, cell density (histograms) and quality-flagged bins (<= 3 cells; potentially less reliable), heatmaps for modules M1ortho-M3ortho, and author-provided annotations. Left side annotation bars show module membership of each gene orthologue. **e, f** Average cross-correlation lag between mouse and human datasets (from panels **c, d**) and the corresponding zebrafish gene.

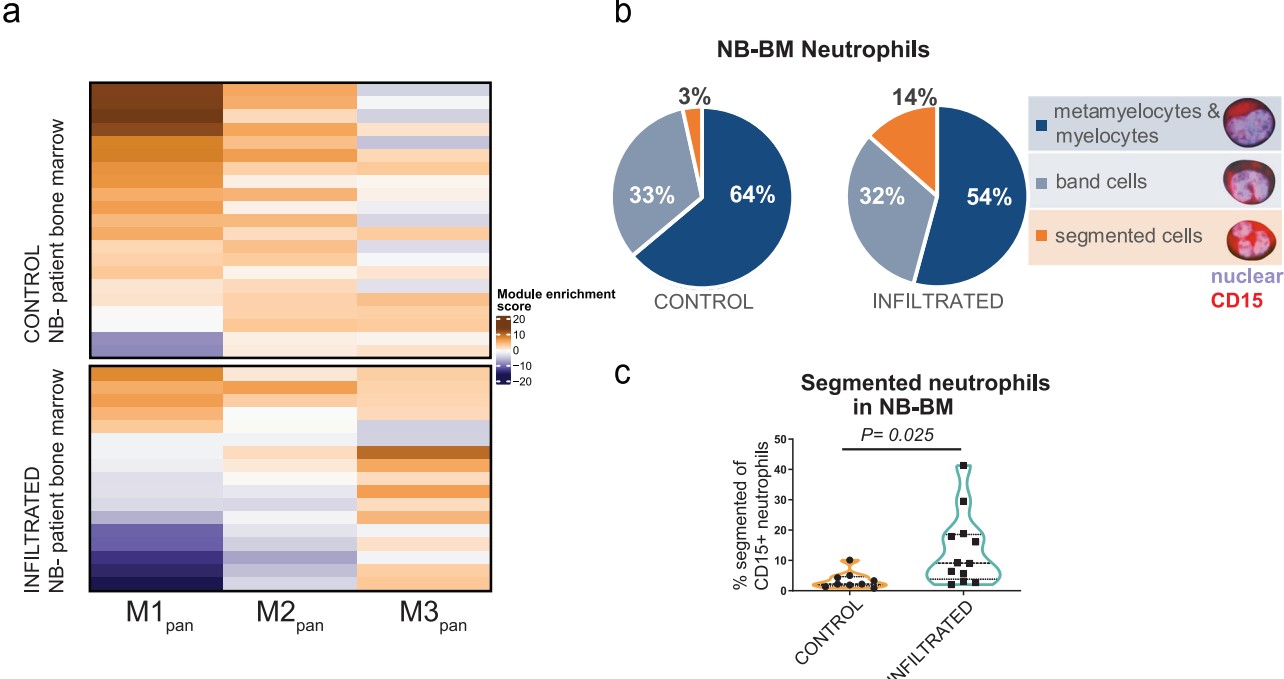

**Fig. 8 | A pan-species neutrophil maturation signature shows the enrichment of mature neutrophils in BM of metastatic neuroblastoma patients. a** Application of the pan-species neutrophil maturation signatures (M1pan, M2pan, M3pan) derived from maturing zebrafish neutrophils (see *Methods* for details on signature definition) on bulk RNA sequencing data from 38 bone marrow (NB-BM) samples of patients with metastatic ($n = 17$, infiltrated) and localized ($n = 21$, control) neuroblastoma. The heatmap displays the ssGSEA[57] score of the maturation signatures on RNA-seq samples from 38 neuroblastoma patients with ($n = 17$) and without ($n = 21$) tumor cell bone marrow infiltration. **b, c** Assessment of neutrophil maturation based on nuclear morphology in BM cytospin samples from patients with metastatic ($n = 12$, infiltrated) and localized ($n = 9$, control) neuroblastoma. Samples were stained with DNA-intercalator Iridium and anti-CD15-Bi209, analyzed with IMC and counted blinded. **c** Analysis of the frequency of segmented neutrophils in control NB-BM (mean = 3.48%; range = 0.8–10.1%) versus infiltrated (mean = 13.48%; range 2–41.3%, two-tailed unpaired *t* test, $P = 0.025$).

release from the marrow[12]. Additionally, we also discovered that *cebpb* expression was required to drive tertiary granule genes such as *mmp9* and *fcer1gl*[61] and for the generation of Mmp9+ neutrophils, suggesting a role beyond emergency granulopoiesis[46,47] that may be underappreciated in other organisms. Another family member, *cebpe* is known to regulate early neutrophil maturation in mice[9] and its putative zebrafish orthologue *cebp1*[62] has been reported to regulate granule genes *lyz* and *srgn* in zebrafish[63]. Indeed, *lyz* and *srgn* were also expressed in early neutrophil maturation in our study (Fig. 4) and *cebp1* itself was weakly expressed throughout maturation. Given that there were no strong changes in *cebp1* expression during maturation, our algorithms prioritized other well-known hematopoietic regulators such as *myb*[64], *myca*, and *ybx1*[45] as driving the early phase of development in our analyses.

Our cross-species comparison identified a subset of genes with highly synchronous expression dynamics in zebrafish, mice, and humans (M1pan-M3pan). We found this conserved gene signature helpful in interpreting neutrophil states also in human bulk RNA sequencing data, as illustrated here using data from BM metastases of pediatric neuroblastoma patients[56] (Supplementary Fig. 17).

Intriguingly, we detected the mature neutrophil signature (M3pan) in patients with disseminated tumor cells. We speculate that this enrichment in mature neutrophils could either result from prolonged retention of mature TANs in the BM or increased recruitment of peripheral neutrophils in response to metastasizing tumor cells. The presence of TANs in BM metastases has been associated with a pro-inflammatory and concurrently immuno-suppressive environment in the BM metastatic niche[56,65]. In the future, our signature could be applied to relate neutrophil maturation grade to disease progression or outcome. However, care must be taken when performing such analysis, as our signatures were designed to identify neutrophil maturation stages and not to distinguish between neutrophils and other cell types. Other cells, in particular closely related myeloid phagocytes, utilize similar genes in their maturation and may influence analyses of bulk tissues. (Please refer to our Supplementary Figs. 10–15 for additional assessments of the specificity and sensitivity of the pan-species neutrophil maturation signatures.) Therefore, we advise validation of putative differences in neutrophil maturation states using orthogonal assays (e.g., using imaging as illustrated in our study).

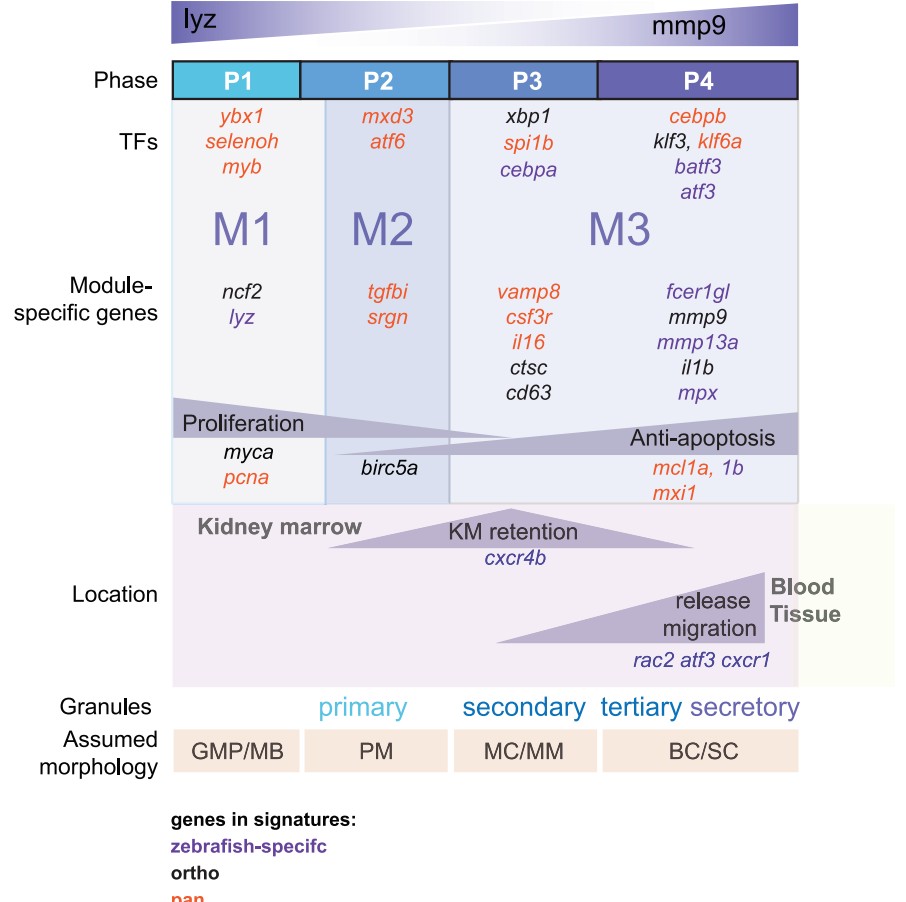

**Fig. 9 | Model for neutrophil development in zebrafish based on scRNA sequencing.** Association of gene expression with modules and phases as determined by scRNA-seq. Granule stage and morphology were assumed by comparing gene expression (as in Fig. 4d, 7a, Supplementary Data 2) with previously published data for neutrophils. TF = transcription factor; module = M; P = phase; KM = kidney marrow; GMP = granulocyte-monocyte progenitor; MB = myeloblast; PM = promyelocyte; MC = myelocyte; MM = metamyelocyte; BC = band cell; SC = segmented cell.

In conclusion, the strong homology of neutrophil maturation supports the translatability of zebrafish models to study neutrophil biology. The combination of live imaging and transcriptomic approaches in zebrafish will further enable the dissection of the role of immature and mature neutrophils in the tumor microenvironment and the consequences of their interactions with tumor cells.

## Methods

### Experimental model and subject details

**Zebrafish model and zebrafish transgenic lines.** Zebrafish (*Danio rerio*) were maintained at standard conditions[66] according to the guidelines of the local authorities (Vienna Magistrat MA58) under licenses GZ:565304/2014/6 and GZ:534619/2014/4 in a research fish facility (Tecniplast, Italy). Experiments on larval zebrafish were carried out at developmental stages, which do not require ethical approval. Single-cell RNAseq or flow cytometry was performed on kidney marrow cells from adult zebrafish post mortem.

Larvae were kept in egg medium with 20 mg/l phenylthiourea (PTU) (Merck) from 22 hpf to avoid pigmentation. Tricaine was used as an anesthetic.

The following transgenic lines were used: *Tg(lysC:CFP-NTR)vi002*,*Tg(lysC:dsRed)nz50Tg*, *Tg(mpeg1:mCherry)gl23*, *Et(kita:GAL4)hzm1*, *Tg(UAS:EGFP-HRAS_G12V)io006*, *Tg(HRAS_G12V:UAS:CFP)vi004*, *Tg(BACmmp9:Citrine-CAAX)vi003*. Transgenic lines were on a AB* (*Tg(lysC:CFP-NTR)vi002*,*Tg(lysC:dsRed)nz50Tg*, *Tg(mpeg1:mCherry)gl23*, *Et(kita:GAL4)hzm1*, *Tg(UAS:EGFP-HRAS_G12V)io006*, *Tg(HRAS_G12V:UAS:CFP)vi004*) or AB*x SAT mixed background (*Tg(BACmmp9:Citrine-CAAX)vi003*). *Tg(BACmmp9:*

*Citrine-CAAX)vi003* fish (abbreviated mmp9:Citrine) were generated by BAC transgenesis according to published protocols[67,68]. In short: Identity of annotated BAC CH211-269M15 (105.8 kb) containing *mmp9* gene and regulatory regions (BACPAC Resources) was confirmed by sequencing, recombineered with iTol2 sites and with membrane-targeted fluorescent Citrine-CAAX DNA inserted at the start codon of *mmp9*. BAC DNA (67 ng/μl) was micro-injected into fertilized zebrafish eggs together with Tol2 transposase mRNA as previously described[69]. pDEST-lysC:CFPNTR (#54) was generated by gateway recombination of p5-lysC (#31), pENTR1A-CFPNTR (#51) and pDEST-Tol2pA2 (kind gift of Chi-Bin Chien). *Tg(HRAS_G12V:UAS:CFP)* plasmid was constructed and kindly provided by the Mione lab. The transgenesis constructs were injected into fertilized zebrafish eggs at 25 ng/μl together with 25 ng/μl Tol2 RNA. Injected zebrafish were grown up to adulthood and screened for germline transmission.

**Generation of Cebpb CRISPR-Cas9 mutant zebrafish.** *Cebpbmut vi006* was generated by microinjection of Cas9 RNP complexes into one-cell stage eggs from *Tg(lysC:CFP-NTR)vi002*x *Tg(BACmmp9:Citrine-CAAX)vi003* crosses. RNPs were assembled by first incubating Alt-R tracrRNA ATTO550 (IDT) (3 μM) with each of ALT-R crRNA dr.*cebpb*_AB and crRNA_ dr.*cebpb* AA (1.5 μM each) in Duplex buffer (IDT) for 5 min at 95 °C followed by mixing equal amounts of assembled gRNA and of Cas9 (0.5 μg/ml) at 37 °C for 10 min. F0 larvae were checked for mutation efficiency by PCR using genotyping primers cebpb_fw (CTGGGCAGGCAACCTATCAC) and cebpb_rev (CATTT-TACCGCCCGCTTGAG) and T7 mismatch assays. Adult F0 and F1 fish

were typed by PCR and sequencing using the aforementioned primers (Supplementary Figs. 6b, 18).

**Human research participants.** In this study bone marrow aspirates from children and adolescents (age at diagnosis 0−18 years) with neuroblastoma or ganglioneuroma were analysed. The cohort used for imaging ($n = 21$) contained samples from 10 female and 11 male patients. Additionally, we collected neuroblastoma bulk RNA-seq data for 38 samples (17 with and 21 without bone marrow infiltration) from accession GSE172184 [Gene Expression Omnibus]". Sex was not included in the parameters for the data analysis results presented.

Patients were recruited previously: Patients with high-risk metastatic (stage M) neuroblastoma were enrolled in the SIOPEN/HR-NBL-1 trial (NCT01704716) according to the trial's inclusion and exclusion criteria. Samples were available as left over material from diagnostic procedures and selected for this study based on their availability in the CCRI Biobank. Ethics oversight by the Ethics committee of the Medical University of Vienna, Austria.

## Method details

**Flow cytometry and FACS.** Single-cell suspensions from adult zebrafish (3 months) spleens or kidneys were prepared by mashing or pipetting, respectively, and pipetting through a cell strainer. To isolate cells from larvae, they were immersed in 10 mM DTT (Merck) in E3 to remove mucus, then digested with Liberase Blendzyme TM at 1.1 µl/ml and Dnase I at 40 µg/ml (Merck) in HBSS under constant shaking at 37 °C for 40 min. 7-AAD (Invitrogen) was used as a live stain. Samples were run on an LSRFortessa cytometer (Becton-Dickinson) with BD FACSDiva Software v9.0 and analyzed in FlowJo_v10.8.1 or sorted using a FACSAria Fusion.

**Wounding, injection of bacteria and phagocytosis assay.** Zebrafish larvae at 2 dpf were wounded in the ventral fin area using a scalpel blade or glass needle and were imaged from 30 mpi (minutes post injury) to 150 mpi on a Leica SP8 confocal microscope. Tracking of LysC+ Mmp9$^{-, INT or HI}$ neutrophils was performed using the ImageJ Fiji TrackMate plugin;[70] The trackmate feature linearity of forward progression = mean straight line speed/ mean speed, where the mean straight line speed is net distance/total track time.

*E. coli* labeled with mCherry (pZS*12-mCherry-KANr: PRlambda-mCherry in pZSstar, SC101* ORI)[71] were injected at OD = 2 into the caudal vein area or otic vesicle as previously described[72]. To study the phagocytic capacity of neutrophils, cells from *Tg(lysC:CFP-NTR)$^{vi002}$/ Tg(BACmmp9:Citrine-CAAX)$^{vi003}$* larvae were analyzed after 6 hpi (hours post infection) by flow cytometry as described above or imaged using confocal microscopy. For ROS detection *E.coli*-mCherry OD = 3 were injected together with CellROX Deep red (Thermo Fisher) at 5 µM into the otic vesicle of 3 dpf larvae.

**Morpholino and mRNA micro-injection.** We used a previously published C/ebp-β translation blocking morpholino (5'-GATCTTAACACCCGCCGGATTGCG-3') and a negative control morpholino (5'-CCTCTTACCTCAGTTACAATTTATA-3') from Gene Tools LCC[46,73]. Efficacious doses (1 pmole and 0.5 pmole, respectively) were determined empirically. Full-length *cebpb* mRNA (50 pg), prepared by mMessage mMachine T7 Ultra transcription (ThermoFisher) or morpholinos were injected into one to two-cell stage embryos from *Tg(lysC:CFP-NTR)$^{vi002}$x Tg(BACmmp9:Citrine-CAAX)$^{vi003}$* crosses.

**Cytospin, Electron microscopy and Sudan Black staining.** Kidney marrow cells of adult *Tg(lysC:dsRed)$^{nz50Tg}$/ Tg(BACmmp9:Citrine-CAAX)$^{vi003}$* fish (aged three months) were FACS sorted into Mmp9$^{+}$LysC$^{+}$ or Mmp9$^{-}$LysC$^{+}$ fractions and spun in a cytocentrifuge (Fisher Scientific) onto slides according to the manufacturer's instructions and stained using Pappenheim solution (Merck).

For TEM Mmp9$^{HI}$LysC$^{+}$ or Mmp9$^{-}$LysC$^{+}$ cells were FACS sorted from adult kidneys of *Tg(lysC:CFP-NTR)$^{vi002}$/ Tg(BACmmp9:Citrine-CAAX)$^{vi003}$* fish and embedded in liquid low melting agarose (3% in PBS), fixed with 2.5% glutaraldehyde in 0.1 M cacodylate buffer containing 5 mM CaCl$_2$, post-fixation in 1% aqueous OsO$_4$ and embedded in Epon812 resin. Semi-thin sections were stained with toluidine blue and images were recorded using a 100x/N.A. 1.4 lens. Ultrathin sections were contrasted with uranyl acetate and lead citrate and imaged using a FEI Tecnai G2 20 TEM. Measurements were performed using Fiji software.

For analysis of granulated neutrophils whole larvae *Tg(lysC:CFP-NTR)$^{vi002}$/ Tg(BACmmp9:Citrine-CAAX)$^{vi003}$* were PTU treated, and stained at 2 dpf with Sudan Black B (Merck) according to a published protocol[74] and imaged on a confocal microscope.

**IMC analysis of neuroblastoma BM cytospin preparations.** Archived cytospin preparations of human BM aspirates from patients with localized and metastatic neuroblastoma (leftover samples from standard-of-care diagnostic procedures) have been obtained after institutional review board approval and informed consent by patients or their guardians from the CCRI Biobank (EK1853/2016). Inclusion criteria: male and female patients aged 0 to 18 years with clinically, histologically, and biologically confirmed high-risk neuroblastoma with bone marrow metastasis or ganglioneuroblastom, ganglioneuroma or localized neuroblastoma with no detectable BM metastasis. Ethical approval for the use of bone marrow aspirates for imaging and clinical data (histological diagnosis) was obtained from the local institutional review board of the Medical University of Vienna (EK1216/2018, EK1224/2020). Patients were not compensated for study participation. Patients' sex was recorded, but not used to disaggregate the data, due to the limited number of patients and lack of statistical power. Gender was not assessed since NB is a tumor of early childhood.

Samples were thawed for 15 min at RT, fixed in 4% PFA (Carl Roth) at 4 °C for 30 min, washed twice in TBS and then blocked with 2% BSA (Carl Roth) and 0.1% Tween-20 (Merck) in TBS at RT for 1 h. Samples were incubated overnight at 4 °C with diluted CD15 antibody (BioLegend; catalog number: 301902; clone: HI98) at 5 µg/ml in 0.1% Tween-20 in TBS. On the following day, samples were stained with the DNA intercalator Iridium (Fluidigm) and then dried with pressured air before IMC measurement with the Hyperion imaging system (Fluidigm) and CyTOF Software v7.

Morphological assessment was subsequently performed in the image analysis software QuPath[75].

Nuclear morphology of CD15$^{+}$ cells was assessed, and cells were classified as segmented cells, band cells or metamyelocytes and myelocytes (3 ROI/patient; $n = 248−330$ cells/patient). Cells were counted using the QuPath counting tool.

**Quantitative real-time PCR.** RNA from FACS sorted cells was prepared using RNeasy Micro kit (Qiagen) and transcribed with High Capacity cDNA transcription kit. qPCR was performed using Maxima SYBR green mix (Applied Biosystems) and gene-specific primers (Supplementary Table 1 in Supplementary Information) on a 7500 Fast real-time PCR machine (Applied Biosystems).

**Imaging.** For imaging larvae were pre-treated in E3/PTU, anaesthetized with 0.02% tricaine and embedded in 1.2% low-melting agarose (Merck) on glass bottom dishes (D35-14-1.5-NJ, Cellvis, USA) as described previously[76]. Images were acquired with a Leica TCS SP8 WLL microscope (HCX PL APO CS 10x/0.40 DRY objective or HC PL APO CS2 40x/1.10 WATER objective). Maximum projections were performed in the Leica LASX 3.7.0.20979 software. Images were rendered using Leica LAS software, Photoshop CS6 (Adobe) and Fiji ImageJ, e.g., for preparing cell tracks. Leukocyte units were analyzed according to[77].

**Single-cell RNA sequencing.** Kidney marrow from 2 adult, six-month old, male fish was isolated, labeled using lipid-tagged indices as published (MULTI-seq)[78]. In short: each kidney marrow was split into four portions, labeled with lipid anchor plus individual barcode solution (2 μM) for 5 min on ice and then incubated with lipid co-anchor (2 μM) for 5 min on ice. Each cell portion was individually FACS-sorted to obtain one population (Mmp9 NO, INT, HI or WKM). All cells were gated on live gate, for WKM debris was excluded in a FSC/SSC gate, and Mmp9 NO, INT, HI were gated on LysC:CFP positivity and different levels of Mmp9:Citrine-CAAX expression (Supplementary Fig. 3a). For multiplexing, the two WKM populations were sorted into one well (A2) and processed together (20,000 cells each; total: 40,000). Similarly, the six LysC+ populations were sorted into one well (A1) and processed together (total: 41,000 cells). Here, cell numbers ranged from 1700 to 10,000 cells/population.

Single cell suspensions were immediately subjected to scRNA-seq using the Chromium Single Cell Controller and Single Cell 3′ Library & Gel Bead Kit v3.1 (10x Genomics, Pleasanton, CA), according to the manufacturer's protocols (10x Genomics). Sequencing was performed at the Biomedical Sequencing Facility of the CeMM Research Center for Molecular Medicine of the Austrian Academy of Sciences (Vienna, Austria) using the Illumina NovaSeq platform and the 50 bp paired-end configuration with adapted read-lengths for both forward and reverse reads.

### Bioinformatics analyses

**Read processing, quality control, and normalization.** We used the CellRanger v3.1.0 software (10x Genomics) for cell demultiplexing and alignment to GRCz11-3.1.0 zebrafish reference transcriptome that had been expanded to include the sequences of reporter genes (*Citrine* and *CFPNTR*, sequences from snapgene.com). The R statistics software v4.0.3 was used to carry out the entire analysis workflow. Processed data was loaded into R for sample demultiplexing (package deMULTIplex[78] v1.0.2; cell classification step was stopped when negative cell number dropped below 100). Only cells classified as singlets or negatives were retained, while doublets were excluded. Next, we loaded the counts into Seurat v4.0.2[79] and performed quality control by only including features detected in at least 20 cells (*n* = 12,619 out of 25,109 features retained), and cells with a minimum of 500 features, mitochondrial reads proportion less than 10%, and doublet score, calculated using function *doubletCells* (package scran[80] v1.18.3;default parameters), below 3 (*n* = 19,373 out of 28,534 cells retained). Remaining negative cells (that is, cells without an assigned sample label; *n* = 4723) from each sequencing run were reclassified separately using Linear Discriminate Analysis (function *lda* from the package MASS v7.3-53; default parameters), only relabeling negative cells with a posterior probability >= 0.95 (*n* = 3500 cells), yielding a final dataset of 18,150 cells. Cell cycle phase inference was done based on expression of G2M and S phase markers saved in Seurat package and using *CellCycleScoring* function from Seurat package. Cells with high level of expression of either G2M or S markers, as they are anti-correlated, are likely to be cycling cells. Next, we normalized raw read counts using SCTransform[81] v0.3.2 and integrated using Harmony v1.0[82] (default parameters) the batch effect of different sequencing runs and fish. We performed low-dimensional projection using UMAP based on the top 30 Harmony components. Reference-based cell annotation using Seurat[79] v4.0.2 was carried out by mapping our data to two atlases of hematopoiesis in zebrafish[24,38] (preprocessing as described above) according to the workflow recommended by the developers (https://satijalab.org/seurat/articles/multimodal_reference_mapping.html; accessed on 18-Nov-2022). Cells identified as neutrophils in both atlases (*n* = 15,876) were selected for downstream analysis. Finally, after subsetting neutrophils, we excluded hemoglobin-related genes (based on the following regular expression vector "^hb[ba]|si:ch211.5k11.8") and features detected in

less than 20 neutrophils to minimize the effect of other cell types on the downstream analysis of neutrophils.

**Inference of neutrophil maturation trajectories, associated genes, and top regulators.** Continuous maturation trajectories were inferred using Slingshot[39] v1.8.0 (default parameters) based on the top two Harmony components of the dataset[82]. We compared Slingshot output to three other methods: Component 1, TSCAN (default parameters), and Monocle3 (use_partition = FALSE, verbose = FALSE, close_loop = FALSE)[83,84]. The first, cluster-free approach uses Harmony 1 coordinate values (akin to principal components), while the other two approaches use pre-defined clusters as inputs (Supplementary Fig. 4e). We then discretized the trajectory into 100 bins by averaging overlapping cells at each bin. We used the functions *fitGAM* (package tradeSeq[40] v1.4.0, parameters: knots = 6, cellWeights = rep(1, #genes)) and *associationTest* (package tradeSeq[40] v1.4.0) to carry out differential expression analysis along the inferred trajectory using pseudotime values from Slingshot. In more detail, first, expression patterns of each gene were modeled as non-linear functions of the pseudotime using generalized additive model (GAM). Second, genes were tested (two-tailed) for the equality of all smoother coefficients within the inferred lineage. We excluded mitochondrial and ribosomal genes, as well as genes with less than 5 cells with 3 or more reads. Genes that passed the cutoff (BH (Benjamini Hochberg)-adjusted $P < 0.05$) were put in descending order based on Wald-statistic and the top 1500 genes were selected as maturation-associated genes for further analysis (Supplementary Data 1). To define gene modules and maturation phases, we used the binned expression of these maturation-associated genes, calculated the DTW distance using function *dist* (package proxy v0.4-25; method = "dtw"), and used hierarchical clustering (parameters: method = "ward.D2"). The resulting dendrograms were cut into three gene modules and four maturation phases, which was the optimal number based on the lowest Kelley-Gardner-Sutcliffe penalty using function *kgs* (package maptree v1.4-7; default parameters) (Supplementary Fig. 16). To interpret and characterize the genes in each maturation module, we used the function *hyper* (package hyper[41] v2.0.1; default parameters) with five datasets (CP:KEGG, GO:BP, CP:WIKIPATHWAYS,CP:REACTOME, HALLMARK) retrieved from Molecular Signatures Database (MSigDB) using *msigdbr* function (package msigdbr v7.5.1; species = "Danio rerio"), to carry out over-representation analysis (upper one-tailed hypergeometric test: FDR (false discovery rate)-adjusted $p < 0.1$; Supplementary Data 3), selected the top five genesets with lowest FDR, and plotted their ratio of overlap with the gene modules. To identify the transcription factors ("top regulators") that best explain the variability in the expression patterns of each gene module, we collected a list of zebrafish transcription factors based on both AnimalTFDB3 database (date of retrieval: 21 June 2021)[85] and manual selection from literature[6,8,12,14], and calculated the DTW distance between each transcription factor and all target genes[85]. DTW allows to account for the expected lag between transcription factor expression and target gene activation. Next, we transformed the DTW distance $d_{il}$ for gene $i$ and transcription factor $l$ into a "scaled DTW similarity" $s_{il}$, as follows: $s_{il} = 1 - (d_{il} / max(d_i))$. Finally, to rank putative regulators per module, we used the function *dunn_test* (package rstatix v0.6.0; default parameters: p.adjust.method = "holm") to run a post-hoc Kruskal-Wallis test (two-tailed) of the distribution of similarity values between transcription factors and putative target genes across the three modules (Supplementary Data 5).

**Cross-species integration and comparison.** For maturation staging of the zebrafish phases, we downloaded relevant datasets[13,48–50] (see Data Availability section) and selected neutrophil-related samples. Samples were then mean-aggregated per maturation stage followed by quantile normalization across all datasets. Specifically, genes were first ranked per column and sorted ascendingly. Next, we replaced each

gene with its mean rank and restored the original ranking of genes. This was followed by scaling quantile-normalized genes across the maturation stages of each dataset. We then used hierarchical clustering (parameters: method = "ward.D2") to group maturation stages across all datasets based on the Pearson distance. Finally, each stage was assigned a continuous score reflecting its maturation order (earliest = 0 and latest = 1) in the corresponding study and the leaves of the resulting hierarchical clustering tree were then reordered under the constraints of the tree using function *reorder* (package: stats, parameters: agglo.FUN = mean).

For gene-level alignment, scRNA-seq[10,12,14,52–54] (see Data Availability section) data were preprocessed following the same approach described under *Read processing, quality control, and normalization* section, with the following exceptions: We did not carry out further quality control filtration on data from Suo 2022[52] (minimum of 501 features and 2001 UMI counts per cell) and Tabula Sapiens[53] (minimum of 200 features and 2500 UMI counts per cell). In the case of neutrotime[12] model, we used the pre-defined embedding provided by the authors via ImmGen single cell explorer (https://singlecell.broadinstitute.org/single_cell/study/SCP1019/ly6-neutrophils-from-bone-marrow-blood-spleen?scpbr=immunological-genome-project#study-download).

We then selected only healthy bone-marrow neutrophils (using provided cell annotations) (Supplementary Data 8). Subsequently, we carried out trajectory inference using Slingshot based on the top two PCA components and discretization into 100 bins, as explained above. Next, we performed homology mapping across zebrafish, mouse, and human using *getLDS* function (package biomaRt v2.46.3; default parameters). Common genes from human and mouse were renamed to the corresponding homologous zebrafish gene using a modified version of *RenameGenesSeurat* function (package Seurat.utils v1.4.7; https://doi.org/10.5281/zenodo.7228243). Cases of 1-to-many homology were resolved by mapping the genes to the most expressed homolog based on the attribute "detection_rate" from the sctransform model. We kept only the set of genes that are common across all datasets, scaled each of the datasets separately, and ordered the genes based on their expression pattern across all datasets using the *seriate* function (seriation 1.3.1; method = "PCA") and the corresponding module. Finally, we used cross-correlation to align the gene expression pattern along maturation trajectory in mouse and human datasets onto zebrafish using the *ccf* function (stats v4.0.3; default parameters) and recorded the mean lag at which highest cross-correlation was achieved among mouse and human datasets per gene (Supplementary Data 7).

The pan-species neutrophil maturation signatures (M1$_{pan}$, M2$_{pan}$, M3$_{pan}$) were defined based on two inclusion criteria: First, consistent expression pattern between human and zebrafish (|maximum cross-correlation lag| <= 50). Second, we excluded unique known markers of other cell types (Additional file 5 from xCell[86]). We then obtained normalized count data from neuroblastoma bulk RNA-seq[56] and used this signature to carry out single-sample gene set enrichment analysis (ssGSEA) using function *ssgsea* (package corto[57] v1.1.11;default parameters). To examine the specificity of neutrophil maturation modules, we calculated module scores for M1$_{pan}$ and M3$_{pan}$ across different cell types and cell states. First, we used two zebrafish scRNA-seq datasets[24,38] and computed module score using *AddModuleScore* function (package Seurat v4.0.2). Second, we used two human scRNA-seq datasets[52,53] that we aggregated into pseudobulk samples of cell types across tissues (>= least 50 neutrophils cells) and one bulk RNA-seq dataset[51]. The human bulk/pseudobulk datasets were then normalized using *vst* function (package DESeq2 v 1.30.0)[87] and a module score was computed using *gsva* function (package GSVA v 1.38.2; method = "ssgsea"). Sensitivity analysis was carried out by mixing cells from the three neutrophil maturation stages (promyelocyte,

myelocyte, neutrophil) at defined ratios (0%, …, 100%) with stromal cells from different tissues[52] to create pseudo-bulk samples with known neutrophil contributions. The mixing was done via random sampling of cells to a sum of 1000 cells and the process was repeated 20 times. Module scores were computed using *gsva* function (package GSVA v 1.38.2; method = "ssgsea"), and the median value presented in Supplementary Fig. 15.

## Quantification and statistical analysis

GraphPad Prism Version 8.3.0 was used for statistical analysis. Statistical details of experiments can be found in the results, methods and figure legends.

## Resource availability

Further information and requests for resources and reagents should be directed to and will be fulfilled by the lead contact, Martin Distel (martin.distel@ccri.at). Raw data underlying the figures will be shared upon reasonable request.

## Material availability

Transgenic zebrafish lines generated in this study are deposited in the European Zebrafish Resource Center (EZRC) (https://www.ezrc.kit.edu/), *Tg(lyz:CFP-NTR)*$^{vi002}$ #37873 and *TgBAC(mmp9:Citrine-CAAX)*$^{vi003}$ #37874.

## Reporting summary

Further information on research design is available in the Nature Portfolio Reporting Summary linked to this article.

## Data availability

The single-cell RNA sequencing data generated in this study was deposited in the Gene Expression Omnibus (GEO) database under accession code GSE252788. The two samples generated are accessible under GSM8007849 (MF317_A1_GEX_zebrafish_multiseq) and GSM8007850 (MF317_A2_GEX_zebrafish_multiseq). We made use of the following publicly available datasets in our study: Single-cell RNA-seq: E-GEOD-100911, E-MTAB-5530, GSE137539, GSE165276, GSE149938, GSE142754, fetal-immune (https://developmental.cellatlas.io/fetal-immune), Tabula sapiens (https://cellxgene.cziscience.com/collections/e5f58829-1a66-40b5-a624-9046778e74f5), Murine neutropenia (https://www.synapse.org/#!Synapse:syn16816566) ; Bulk RNA-seq: GSE79044, GSE109467, GSE172184 neuroblastoma data provided by the authors (https://www.ncbi.nlm.nih.gov/geo/query/acc.cgi?acc=GSE172184), GSE153263, GSE151682, GSE175880 ; Others: neutrotime model (https://singlecell.broadinstitute.org/single_cell/study/SCP1019/ly6-neutrophils-from-bone-marrow-blood-spleen?scpbr=immunological-genome-project), AnimalTFDB3.0-zebrafish (http://bioinfo.life.hust.edu.cn/static/AnimalTFDB3/download/Danio_rerio_TF), xCell signatures Additional file 5 (https://static-content.springer.com/esm/art%3A10.1186%2Fs13059-017-1349-1/MediaObjects/13059_2017_1349_MOESM5_ESM.xlsx), additional metadata of GSE109467 was kindly provided by the authors. Source data are provided with this paper.

## Code availability

Computer code used for the data analysis in this paper can be accessed on our GitHub page https://github.com/cancerbits/Kirchberger_Shoeb2024_neut and via Zenodo.

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

## Acknowledgements

We would like to thank Dieter Printz for FACS sorting; the Biomedical Sequencing Facility of the CeMM Research Center for Molecular Medicine of the Austrian Academy of Sciences for next-generation sequencing; Cristina Santoriello and Marina Mione (University of Trento) for the kind gift of the *HRAS_G12V:UAS:CFP* plasmid. We would like to acknowledge the CCRI Biobank and Marie Bernkopf. Peter Repiscak for providing support on pre-processing of NB-BM sequencing data. We would like to thank the members of the Innovative Cancer Models group and Dietmar Herndler-Brandstetter for their insightful comments on the manuscript; Adam Varady for providing illustrations; Ulrike Pötschger for advice on statistical tests. This study was supported by the St. Anna Kinderkrebsforschung (to S.T.M., R.L., F.H. and M.D.), the Austrian Research Promotion Agency (FFG) (project 7940628, Danio4Can to M.D.), a German Academic Exchange Service postdoctoral fellowship and an EMBO fellowship (to M.D.), the Austrian Science Fund (FWF) through grants TAI454 (to F.H. and M.D.), TAI732 (to F.H.), I4162 (ERA-NET/Transcan-2 LIQUIDHOPE; to S.T.M.), P35841 (MAPMET; to S.T.M.), P34152 (to T.L.), P30642 (to C.S.) and the Alex's Lemonade Stand Foundation for Childhood Cancer 20-17258 (to F.H. and M.D.), and the Swiss Government Excellence Scholarship (to D.L.), and the EC H2020 grant no. 826494 (PRIMAGE; to R.L.), and by the European Commission within the FP7 Framework program (Fungitect-Grant No 602125 to T.L.).

## Author contributions

SK designed and performed experiments, analyzed data and wrote the manuscript. MS performed bioinformatics analysis including collection and preprocessing of public data, and wrote the manuscript. DL performed IMC on neuroblastoma BM samples. KF supported BAC screening and AWW cebpb[mut vi006] genotyping. LS performed the 10x Genomics workflow. FR designed the graphical abstract. FR and EB developed new imaging methods. RL has provided patient samples and clinical data. BB provided access to IMC instruments. FN, TL and DT provided reagents. CS performed and analyzed electron microscopy imaging. MF supervised scRNA-seq analysis. STM and BB conceptualized and supervised the analysis of human BM samples. FH and MD equally contributed to the manuscript. They designed and analyzed experiments, supervised all work, and wrote the manuscript. All authors contributed to writing the manuscript.

## Competing interests

The authors declare no competing interests.

## Additional information

[1]St. Anna Children's Cancer Research Institute (CCRI), Vienna, Austria. [2]Medical University of Vienna, Department of Dermatology, Vienna, Austria. [3]Labdia - Labordiagnostik GmbH, Vienna, Austria. [4]Medical University of Vienna, Center for Medical Biochemistry, Max Perutz Labs, Campus Vienna Biocenter, Vienna, Austria. [5]Department of Quantitative Biomedicine, University of Zurich, Zurich, Switzerland. [6]Institute of Molecular Health Sciences, ETH Zurich, Zürich, Switzerland. [7]Medical University of Vienna, Department of Pediatrics, Vienna, Austria. [8]Cell and Developmental Biology, University of California, San Diego, CA, USA. [9]Medical University of Vienna, Division of Cell and Developmental Biology, Center for Anatomy and Cell Biology, Vienna, Austria. [10]These authors contributed equally: Stefanie Kirchberger, Mohamed R. Shoeb. [11]These authors jointly supervised this work: Florian Halbritter, Martin Distel ✉e-mail: stefanie.kirchberger@ccri.at; florian.halbritter@ccri.at; martin.distel@ccri.at

