## [Peer Review File · Nature Communications]

Comparative transcriptomics coupled to developmental grading via transgenic zebrafish reporter strains identifies conserved features in neutrophil maturationREVIEWER COMMENTS

Reviewer #1 (Remarks to the Author):

In this manuscript, Kirchberger et al. identified a mature neutrophil marker, *mmp9*, and generated a transgenic line to separate immature and mature neutrophils in zebrafish. They found *cebpb* as a critical transcription factor for generation of mature *mmp9*⁺ neutrophils. The cross-species comparison among human, mouse and fish revealed the conserved gene signatures for neutrophil staging, at least at early neutrophil maturation, which is innovative. The experiments are well characterized, and the data are well presented. However, some concerns should be addressed, as the major and minor points listed below.

Major points:

1. In Fig. 1A, cells at high magnification were presented. The co-localizations of *mmp9* and *lyz*, *mmp9* and *mpeg1* in whole mount fish level should be shown, too. It is necessary to record their colocalization or exclusion percentages of the single or double positive cells. In supplementary Fig. 1, the authors showed the *mmp9*⁺ fluorescent pattern at 2 dpf and 5 dpf, but it doesn't look like the typical dotted pattern of neutrophils. Are the fluorescent signals restricted in neutrophils in early stages? The authors should show more representative images. To confirm the citrine protein correctly represent the transcript of *mmp9*, the authors should combine WISH and antibody staining to show the co-localization of *mmp9* mRNA and Citrine protein.
2. In Fig. 1B, the authors said that no detectable *mmp9* expression in *mpeg:mCherry*⁺ macrophages, but why 35% of *Mpeg*⁺*LysC*⁻ cells still express *mmp9*:Citrine, almost identical to *LysC*⁺*Mpeg*⁻ cells? Besides, why call the Q2 cluster as *LysC*⁺*Mpeg*^{lo} cells but not *LysC*⁺*Mpeg*⁺ cells? These points should be clarified.
3. In Fig. 3A, are those cells labelled with * in the cluster neutrophils or other cell types? Why is the cluster separated from the main neutrophil cluster, which might mean neutrophil maturation is not a continuous process? Or else, the UMAP parameters should be adjusted to avoid this misleading.
4. In Fig. 5B, what does the Y-axis mean? The peaks of M1 and M2 target modules of *cebpb* seemed to be much higher than M3, what does it suggest? It should be clarified.
5. For Fig. 5E, 5F, 5H, 5I, and 5J, the authors should always show the representative images of the zebrafish larvae to make it more convincing. Besides, the n values of Fig. 5F seems too low, does each dot represent multiple larvae? The n values should be clearly indicated.
6. Nowadays, only MO-knockdown data is not well acceptable for characterizing gene function. If the authors could not provide KO data, they should at least verify that the MO efficiently blocked *Cebpb* expression. The authors should also combine *cebpb* overexpression with MO and check whether the MO-knockdown phenotype could be rescued.
7. In Fig. 7A, the authors detected the expression levels of M1, M2, and M3 modules in the bulk RNA-Seq data of BM. Are these modules neutrophil-specific? Do these modules represent the neutrophil patterns in BM only? Are they able to be used for distinguishing neutrophil lineage status changes from comprehensive surrounding tissues? It will be helpful to perform an RNA-Seq analysis with their pre-neoplastic melanoma fish and check if the modules could be widely applied to reflect neutrophil patterns in another tumor types or in other hematopoietic tissues, but not only in BM of neuroblastoma.

Minor points:

1. In Fig. 2C, how is linearity calculated? Based on Line 128-130, it seemed that lower linearity represented more straightforward moving pattern, which is confusing.

2. Page 8 Line 190, *cxcr4b* seems up-regulated but not down-regulated in P3. Please modify the description.
3. In Fig. 4D, Fig. 5A-C, and Fig. 8, the zebrafish genes were not written in italic.

Reviewer #2 (Remarks to the Author):

This is an interesting manuscript that attempts to link MMP9 expression on zebrafish neutrophils with neutrophil maturation. The authors also present a comprehensive comparison of zebrafish, mouse, and human neutrophil transcriptomes. The work represents a significant and rather elegant effort with interesting outcomes. However, I feel that there are some shortcomings in this manuscript that should be addressed.

I am not sure that I agree with the authors' conclusion that *mmp9* expression, as based on the results in Fig.2, correlates with neutrophil maturation. It could be argued that the expression of something like an MMP would be upregulated in situations where a cell type like a neutrophil must extravasate into/migrate through tissues, such as during wound response, infection and/or in the presence of a tumor. There is also a breadth of data suggesting that tumors will elicit MMP expression on myeloid cells in order to facilitate metastasis.

I would argue that neutrophil activation/polarization is not synonymous with differentiation. As an extension of this argument, it is then maybe not adequate to use MMP9 expression as an indicator of neutrophil maturation in their single cell analyses. These results hinge on an unsupported assumption that the level of MMP9 expression tracks with maturation, which in my opinion has not been shown in this paper.

Could the authors demonstrate how MMP9 expression tracks with expression of known markers of neutrophil maturation, such as *C/ebp* transcription factors, in *LysC+* cells? I do not think that it is sufficient to confirm that something like *C/ebpb* is expressed in MMP9+ but not MMP9- cells. Indeed, if as in mammals, *C/ebp* beta expression is associated with neutrophil maturation, you would expect only matured neutrophils to possess MMP expression. However, if as in mammals, MMP expression is also myeloid cell activation state-dependent, you would also have mature myeloid cells that may not have upregulated expression of a given MMP, depending on their activation, rather than differentiation state. Correlation is not causation, so it could be argued that although the authors see MMP9 expression by cells bearing markers of neutrophil maturation, this does not indicate that MMP9 expression is a reliable marker of neutrophil maturation. Undoubtedly, other populations of fully differentiated fish neutrophils have no/low MMP9 expression.

Line 174: It is not clear from the text or the figure legend how the authors are able to deduce the cell cycle stage of the sorted cells. Please clarify.

It is not clear to me how the analyses of fish neutrophils in Figs. 4-6 relate to the neutrophil populations sorted based on MMP9 expression and depicted in Fig. 3. Please clarify and maybe find a way to indicate that in your figures/legends.

The cross-species comparisons are very interesting and valuable to the field.

Reviewer #3 (Remarks to the Author):

This article generates zebrafish models for different mature stages of neutrophil. Through single cell RNA-seq analysis, the authors discovered four different phases of neutrophil and three gene modules and transcription factor regulators associated with these stages in Zebrafish. By clustering module genes in zebrafish, human, and mouse samples, the expression pattern is similar across

species in each stage. These genes also show similar trends along mature stages in zebrafish, human, and mouse. The lag in expression is smaller for early-stage module genes and larger for late-stage module genes across species. Known immature neutrophil gene signatures are more in late-stage gene modules. The resulting gene modules in this article can be an addition to the existing gene signatures. They can differentiate different maturation stages as well as can be a pan-species gene signatures shared among zebrafish, human, and mouse. The application of the signatures to human bulk RNA-seq data shows proper stage estimation of each sample. The discovered gene modules provide a robust alternative to existing neutrophil maturation signature in identifying neutrophil maturation stages from gene expression data. The work is well designed and provides sufficient evidence and explanation.

The tissues of the samples used should be explicitly presented. The effects of tissue used on derivation of gene modules and identification of neutrophil maturation stage should be discussed.

The authors use slingshot to infer trajectory. As it is prediction method and some of the conclusions are relying on the results of this analysis, it will be more robust to double check with another inference application.

"transcriptional mechanisms" is mentioned in the title, while few contents are related, thus should be avoided.

Response to reviewer comments

We would like to thank the reviewers for their insightful comments, which helped to further improve our manuscript. We will address all points of criticism one by one below.

REVIEWER COMMENTS

Reviewer #1 (Remarks to the Author):

“In this manuscript, Kirchberger et al. identified a mature neutrophil marker, mmp9, and generated a transgenic line to separate immature and mature neutrophils in zebrafish. They found cebpb as a critical transcription factor for generation of mature mmp9+ neutrophils. The cross-species comparison among human, mouse and fish revealed the conserved gene signatures for neutrophil staging, at least at early neutrophil maturation, which is innovative. The experiments are well characterized, and the data are well presented. However, some concerns should be addressed, as the major and minor points listed below. “

We are pleased that the reviewer acknowledges the value and novelty of our work and the thoroughness of the presented analyses. Thank you for your insightful critique, which we addressed in the revised manuscript with additional experiments and clarifications in the text.

Major points:

“1. In Fig. 1A, cells at high magnification were presented. The co-localizations of mmp9 and lyz, mmp9 and mpeg1 in whole mount fish level should be shown, too. It is necessary to record their colocalization or exclusion percentages of the single or double positive cells. In supplementary Fig. 1, the authors showed the mmp9+ fluorescent pattern at 2 dpf and 5 dpf, but it doesn't look like the typical dotted pattern of neutrophils. Are the fluorescent signals restricted in neutrophils in early stages? The authors should show more representative images. To confirm the citrine protein correctly represent the transcript of mmp9, the authors should combine WISH and antibody staining to show the co-localization of mmp9 mRNA and Citrine protein.“

Following the reviewer's suggestion, we now included more overview images, insets (**Figure 1a** and **Supplementary Figure 1a**), and representative images of transgenic zebrafish (**Supplementary figure 1c**) to depict the co-localization of Lyz and Mmp9 expression in a subset of neutrophils. Furthermore, we added the co-localization percentages of Mmp9/Lyz/Mpeg single-, double-, and triple-positive cell populations based on analysis of these images (**Supplementary Figure 1c**). To clarify, Mmp9 is not exclusively expressed in neutrophils but is also detected in epithelial cells and the distal gut as shown in Supplementary Figure 1a. Our new co-localization quantification of Mmp9/Lyz/Mpeg in different areas (head, CHT, tail) revealed a stronger overlap of Lyz and Mmp9 in the head region compared to the CHT.

We added the following paragraph to the manuscript and the respective data in Figure 1 and Supplementary Figure 1 as shown below:

“To examine the identity of Mmp9+ leukocytes, mmp9:Citrine fish were crossed with fish double-transgenic for myeloid markers Tg(lysC:CFP-NTR)^{vi002} (labelling neutrophils), and Tg(mpeg:mCherry)^{gl23}

(macrophages)³¹. Live imaging of triple-transgenic larvae at 3 and 5 dpf confirmed the existence of both *Mmp9*⁻ and *Mmp9*⁺ subpopulations of *lysC:CFP*⁺ neutrophils with a more pronounced co-localization in the head than in the caudal hematopoietic tissue (CHT) region (**Fig. 1a**). *Mmp9*⁺*LysC*⁺ cells had low *mpeg:mCherry* expression, while bona-fide macrophages were highly positive for *mpeg:mCherry* and few of the latter showed detectable *mmp9* expression (**Fig. 1b**). *LysC*⁺*Mpeg*⁺*Mmp9*⁺ triple-positive neutrophils were enriched in the head region at 3 dpf (mean=36%; range=24.3-52.3) as well as in the head and tail at 5 dpf (mean=37%; range=29-46.8; and 32%, range=20.5-40, respectively), whereas the CHT population decreased from 3 to 5 dpf (from mean= 20, range= 11.1-30.3 to 13%, range= 10-17.6) (**Supplementary Fig. 1c**). Pre-dominant localization of triple-positive neutrophils in the head points towards an already more progressed maturation stage of RBI-derived head neutrophils compared to HSC-derived CHT neutrophils. Interestingly, a recent publication also described transcriptional differences between these two neutrophil populations³². “

Fig 1a:

Fig. 1: *mmp9:Citrine* identifies mature neutrophils. a Confocal images of triple transgenic larvae *Tg(lysC:CFP-NTR)^{vi002} Tg(BACmmp9:Citrine-CAAX)^{vi003} Tg(mpeg:mCherry)^{g123}* reveal myeloid cells with different expression levels of the three analyzed markers. Stitched whole-mount images of a 5 dpf larva with insets in the head and CHT regions.

Supplementary Fig. 1a-c:

Supplementary Figure 1a-c: Newly generated BAC transgenic line $Tg(BACmmp9:Citrine-CAAX)^{vi003}$ reporting epithelial and leukocyte expression. **a** $Tg(BACmmp9:Citrine-CAAX)^{vi003}$ zebrafish exhibit transcriptional activity of the *mmp9* locus in epithelial cells of the skin and gut and in dispersed cells. F1 *mmp9:Citrine* larvae were analyzed using fluorescence confocal microscopy on a Leica SP8 with a HCX PL APO CS 10x/0.40 DRY objective for Citrine (in yellow) at 2 and 5 dpf. To cover the whole larvae three individual pictures were taken and stitched. Scale bars 250 µm. Insets show expression in myeloid cells, the epithelia of the distal gut and the tail fin. **b** *Mmp9* RNA is enriched in whole kidney marrow cells from $Tg(lysC:CFP-NTR)^{vi002}/Tg(BACmmp9:Citrine-CAAX)^{vi003}$ double transgenic adult fish sorted for Citrine expression. Kidneys were isolated and FACS sorted into CFP+ Citrine- or CFP+ Citrine+ cells, which were analyzed by quantitative real-time PCR with primers for *mmp9*. $n = 3$, paired t-test $P = 0.0144$. **c** Confocal analyses of co-localizations of LysC, *Mmp9* and *Mpeg* in triple transgenic larvae $Tg(lysC:CFP-NTR)^{vi002} Tg(BACmmp9:Citrine-CAAX)^{vi003} Tg(mpeg:mCherry)^{gl23}$ at 3 and 5 dpf in the head, CHT or tail region. Venn diagrams showing average percentages ($n = 3-7$ larvae) and representative images at 5 dpf (right). Dashed lines indicate pigments. M, a typical *Mpeg*⁺ macrophage; N^T, a *LysC*⁺*Mmp9*⁺*Mpeg*⁺ neutrophil; N^D, a *LysC*⁺*Mmp9*⁺*Mpeg*⁺ neutrophil. Acquired at a Leica TCS SP8 WLL microscope (HC PL APO CS2 40x/1.10 WATER objective).

The reviewer is also asking to verify that Citrine protein correctly localizes to cells expressing *mmp9*. Our verification of this was to sort for Citrine positive neutrophils (protein level) and to confirm by qPCR that these cells express *mmp9* (mRNA level) at significantly higher levels than Citrine negative cells (**Supplementary Figure 1b**), which was indeed the case. (Our additional attempt of a combined WISH for *mmp9* and antibody staining for Citrine was unfortunately not successful and would require more time for further technical refinement.)

2. In Fig. 1B, the authors said that no detectable *mmp9* expression in *mpeg:mCherry*⁺ macrophages, but why 35% of *Mpeg*⁺*LysC*⁻ cells still express *mmp9*:Citrine, almost identical to *LysC*⁺*Mpeg*⁻ cells? Besides, why call the Q2 cluster as *LysC*⁺*Mpeg*^{lo} cells but not *LysC*⁺*Mpeg*⁺ cells? These points should be clarified.

The reviewer's questions made us realize that our previous presentation of Figure 1b (now **Fig 1c**) was misleading. We are now showing new flow cytometry experiments including the controls used for gating to demonstrate that only very few *Mpeg*⁺ cells show weak Citrine fluorescence (**Figure 1c**). Our detailed confocal analysis also yielded a low number of *Mpeg*⁺/*Mmp9*⁺ cells at 5 dpf (5-6% of all *Mpeg*⁺, *LysC*⁺ or *Mmp9*⁺) (see new **Supplementary figure 1c**).

We had initially called cells in Q2 "*Mpeg*^{lo}" because these cells show lower fluorescence intensity for mCherry compared to cells in Q3. To avoid any ambiguity, we have now changed *Mpeg*^{lo} to *Mpeg*⁺ in the text and figures.

We added the following paragraph to the manuscript and the respective data in Figure 1 and Supplementary Figure 1 as shown above:

“Complementary analysis by flow cytometry revealed that 15.4% (mean; range: 12.4- 17.1%) of myeloid cells were expressing Mmp9; of those 67% (mean; range 61.8- 69.5%; Q2) belonged to the LysC⁺Mpeg⁺ neutrophil population (Fig. 1c). Fewer Citrine-positive cells were part of the LysC⁺Mpeg⁻ neutrophil (mean= 29.06%; range: 25.8-34.8% Q1) and Mpeg⁺LysC⁻ macrophage (mean= 3.9%; range 2.8- 5.8%; Q3) populations.”

3. In Fig. 3A, are those cells labelled with * in the cluster neutrophils or other cell types? Why is the cluster separated from the main neutrophil cluster, which might mean neutrophil maturation is not a continuous process? Or else, the UMAP parameters should be adjusted to avoid this misleading.

The cluster in the UMAP labeled by the asterisk (*) consists of sorted neutrophils (depicted in pinkish colors in **Figure 3b** and other hematopoietic cells in grey), which are proliferating (mostly in G2/M phase, cp. **Supplementary Figure 3e**) and hence form a separate cluster. We apologize for not making this clear enough in the original manuscript. We have now added additional figure panels and explanations in the figure legend to make the differences explicit (new **Supplementary Figures 3k-m**). As cells divide, the corresponding transcriptional machinery becomes dominant in the gene expression

patterns, which places the cells “away” from other neutrophils and possibly together with other dividing cells. This is the nature of the non-linear low-dimensional projection in the UMAP, which captures the most powerful expression patterns, but does not imply any discontinuity. Therefore, great care must be taken when interpreting putatively gradual processes (differentiation, maturation) in UMAPs or other low-dimensional visualizations – see Chari & Pachter 2023 (<https://doi.org/10.1371/journal.pcbi.1011288>) for a critical assessment of the use of UMAPs in single-cell genomics. For our trajectory analysis, we therefore worked with principal component analysis (corrected for batch effects with the Harmony algorithm) and algorithms that are independent of the UMAP projection (Slingshot). Furthermore, we show that geneset enrichment analysis of the marker genes of the * labelled cluster (Cluster 3) identified cell cycle-related pathways as top hits, supporting the notion that these cells are dominated by cell-cycle effect and explains the observed separation (Supplementary Fig. 3m).

4. In Fig. 5B, what does the Y-axis mean? The peaks of M1 and M2 target modules of *cebpb* seemed to be much higher than M3, what does it suggest? It should be clarified.

The y-axis in the plots in **Figure 5b** indicates a kernel density estimate – roughly an equivalent of a histogram for continuous values. It therefore indicates for each shown transcription factor (TF) the proportion of genes in the respective maturation gene signature (M1, M2, M3) with a given specificity score on the x-axis. This score is a scaled similarity measure based on dynamic time warping (DTW) distance between a target gene and a TF, with respect to all other TFs (see *Inference of neutrophil maturation trajectories, associated genes, and top regulators* section under Methods for the formula and more details). In other words, we reason that TFs with high specificity (shown as shifted to the right in Figure 4) to many target genes from one of the modules are candidate regulators of this module (which needs to be validated subsequently, as we have for *cebpb* in our manuscript). Our bioinformatics analysis was aimed at quantifying how similar the expression of the genes in each module was to that of each TF (possibly with an offset) and the plots show this similarity for all genes in each module.

We realize the placement of gene labels and lack of axis label were particularly confusing and we have attempted to improve the layout of the figure for clarity (**Figure 5b, Supplementary Figure 5**). We also added an additional explanation to the figure legend and to the manuscript, which reads like this:

„We reasoned that the expression of key regulatory TFs was closely correlated with the expression of target genes, albeit possibly with a temporal delay. We therefore compared transcription factor expression with the expression of genes in each module using dynamic time warping (DTW) analysis, which highlighted *ybx1*, also known as key splicing factor in hematopoietic development⁴⁷, *hmgb2b* and *dnajc1* as top regulators of early maturation genes in module M1 (Fig. 5b-c; Supplementary Fig. 5; Supplementary Table 4, 5).“

5. For Fig. 5E, 5F, 5H, 5I, and 5J, the authors should always show the representative images of the zebrafish larvae to make it more convincing. Besides, the n values of Fig. 5F seems too low, does each dot represent multiple larvae? The n values should be clearly indicated.

We now added representative images for Figure 5e, & 5f. Former Figure 5h actually showed the representative plot for Figure 5i at 5dpf (now **Figure 5h**) and is now included in this figure (**Figure 5**).

Thank you for pointing out that the *n* values were not clearly stated for Figure 5f. Each dot represents a pool of 20-30 zebrafish larvae. We have now more clearly indicated the total number of zebrafish larvae investigated in each experiment shown in Figure 5e-i.

6. Nowadays, only MO-knockdown data is not well acceptable for characterizing gene function. If the authors could not provide KO data, they should at least verify that the MO efficiently blocked *Cebpb* expression. The authors should also combine *cebpb* overexpression with MO and check whether the MO-knockdown phenotype could be rescued.

Initially, we had applied a published and previously validated Morpholino to knock down *cebpb* (Hall et al. 2012; <https://doi.org/10.1016/j.stem.2012.01.007>). Following the reviewer's suggestion, we have now also included data from our recently generated *cebpb* CRISPR mutant to support our previous findings from MO-mediated knock down experiments in zebrafish larvae. Reassuringly, we observed a decreased percentage of *mmp9⁺* cells in the adult *cebpb* mutant under steady-state conditions, confirming a role of C/EBPb in aspects of neutrophil maturation (see new **Figure 5j** and **5k**, and **Supplementary Figure 6** about the generation of the mutant).

We added the following paragraph in the manuscript:

*"To confirm the function of C/ebp-β in adult neutrophils we generated *cebpb*^{MUT vi006} fish by CRISPR-Cas9 introducing a 76 bp deletion plus a 2 bp insertion leading to a frameshift and premature stop codon (Supplementary Fig. 6a,b). In agreement with the results in larvae, we observed a strongly diminished *LysC⁺Mmp9⁺* population in the WKM of homozygous *cebpb*^{MUT vi006} fish (in *Mmp9:Cit^{HOM}**

mean= 1.4%; in *Mmp9:Cit^{HET}* mean=5.1%) compared to *cebpb^{WT}* (in *Mmp9:Cit^{HOM}* mean= 32%; in *Mmp9:Cit^{HET}* mean=25.1%) (Fig. 5j), but no change in the frequency of *LysC⁺* neutrophils (Fig. 5k).”

Furthermore, we add details about the generation and characterization of the mutants in the Methods section.

7. In Fig. 7A, the authors detected the expression levels of M1, M2, and M3 modules in the bulk RNA-Seq data of BM. Are these modules neutrophil-specific? Do these modules represent the neutrophil patterns in BM only? Are they able to be used for distinguishing neutrophil lineage status changes from comprehensive surrounding tissues? It will be helpful to perform an RNA-Seq analysis with their pre-neoplastic melanoma fish and check if the modules could be widely applied to reflect neutrophil patterns in another tumor types or in other hematopoietic tissues, but not only in BM of neuroblastoma.

We thank the reviewer for raising these important points for discussion. Apologies in advance, this will be a lengthy response.

Firstly, it is important to remember that the neutrophil maturation signatures were initially defined to resolve and correctly order the stages of neutrophil maturation and were not defined to distinguish neutrophils from other cell types. Indeed, it is possible that other cell types, especially myeloid cells sharing common developmental origins in the early maturation stages such as monocytes, utilize partially overlapping transcriptional regulators for maturation. We expect the signatures to become increasingly specific in mature neutrophils. Furthermore, for the pan-species neutrophil signatures (*M1_pan*, *M2_pan*, *M3_pan*) we have attempted to exclude known markers of all other cell types to minimize confounding effects of other cell types present in tissues analyzed with bulk RNA-seq.

That being said, we have now performed a detailed assessment of the specificity of our signatures, which we agree with the reviewer will provide important information for the applicability for the analysis of neutrophil maturation stage in different contexts. We performed two types of analysis to assess the specificity and sensitivity of the signatures.

Neutrophil score specificity (“Are these modules neutrophil-specific? Do these modules represent the neutrophil patterns in BM only?”): We utilized existing, pre-annotated single-cell datasets containing many different cell types from different tissues to check the specificity of our maturation signatures. For simplicity, we focused on the *M1_pan* and *M3_pan* signature. We started with the same zebrafish hematopoietic reference datasets that we had already used previously to annotate neutrophils in the paper (Athanasiadis et al. 2017; Tang et al. 2017; cp. **Figure 3c,d**). In both cases, we found that mature “neutrophils and myeloid cells” and cells with neutrophil-linked reporters *lyz* and *mpx* exhibited the strongest *M3_pan* scores while having very low *M1_pan* scores. On the contrary, subsets of HSPCs and/or *runx1* progenitors had a high *M1_pan* score and low *M3_pan* score (**Supplementary Figure 10**). Both datasets also showed (few) neutrophils with medium scores for both signatures, which might represent cells at intermediate maturation stages. Cells of other hematopoietic lineages had a low score for both *M1_pan* and *M3_pan* (e.g., *cd4* / T cells). We then explored the comprehensive cross-tissue immune compartment from the Tabula Sapiens Consortium 2022 dataset. To facilitate the analysis and visualization of this large dataset (>200k cells), we aggregated cells of the same type and

tissues into pseudo-bulk samples. The results display a clear separation of neutrophils from the other cell types while exhibiting the same distinguishing pattern of high *M3_{pan}* and low *M1_{pan}* scores we saw in zebrafish across 10 different tissues (**Supplementary Figure 11**).

Next, we sought to further assess the ability of the two modules to uniquely resolve and order neutrophil maturation stages using the data published by the Teichmann lab on the developing human immune system across organs (Suo et al 2022). We followed the same pseudo-bulking approach previously mentioned to facilitate the analysis and visualization of this large dataset (>500k cells). The mature neutrophil stage (NEUTROPHIL) shows high $M3_{pan}$ score and low $M1_{pan}$ score, the immature stage (PROMYELOCYTE) shows the opposite pattern, and the intermediate stage (MYELOCYTE) is in-between (**Supplementary Figure 12**). In combination, the scores from the two modules resolve and separate the three neutrophil stages across tissues achieving the correct order of maturation (albeit individual scores might be higher, e.g., $M3_{pan}$ for some monocytes). The heatmap shows the rank of maturation stages in neutrophils, monocytes, and macrophages based on $M1_{pan}$ score across tissues (**Supplementary Figure 13**). Neutrophils are the only cell type where the correct order is attained and maintained across tissues.

Finally, we complemented the single-cell datasets we used above with a bulk dataset of myeloid *in vitro* maturation from Ramirez et al 2017. Among limitations of the single-cell dataset that might interfere with our analysis is the imbalance of representation of biological states and cell numbers (e.g., neutrophils in Suo et al 2022 are one and two orders of magnitude fewer than monocytes and macrophages). Thus, using a dataset with controlled experimental conditions and balanced design is of great value to our analysis. The results of this dataset show that differentiating neutrophils (Neu) at early time points had higher M1_{pan} and lower M3_{pan} scores compared to monocytes (Mon) and macrophages (Mac), which gradually switches to the opposite pattern at late differentiation timepoints (**Supplementary Figure 14**). Additionally, while the neutrophil maturation trajectory is correctly recapitulated based on the two modules scores, they completely fail to order the monocyte maturation time points and incorrectly reverse the order of 120 and 96 hours in macrophages.

Neutrophil score sensitivity (“Are they able to be used for distinguishing neutrophil lineage status changes from comprehensive surrounding tissues?”): To evaluate how well maturation status could be assessed in the context of different surrounding tissues, we performed an *in silico* titration experiment in which we mixed in neutrophils at different maturation stages (promyelocyte, myelocyte, neutrophil) and at different ratios (0%, 10%,..., 100%) with stromal cells from different tissues to create pseudo-bulk samples with known neutrophil contributions. We used the same dataset from Suo et al 2022 as in the previous analysis for this purpose. The results show how the modules scores order and resolve the mixtures into increasingly different groups with increased neutrophil percentage (**Supplementary Figure 15**). The distinction between stages can be made at neutrophil percentages as low as 20%. Additionally, this analysis further revealed the role of each module. $M1_{pan}$ score orders and separates the different maturation groups, while $M3_{pan}$ score does a better job at resolving the change in abundance of each maturation group.

Conclusions: Based on our assessments, we believe that we have shown that our neutrophil maturation signatures can indeed be used to explore neutrophil-specific differences in maturity with high sensitivity in the context of different heterogenous bulk tissues. However, it is important to stress that care must be taken in the presence of other hematopoietic cell types and that it is necessary to validate findings (as for any bioinformatics prediction) with orthogonal experiments, such as the imaging data that we utilized for our analysis of neuroblastoma bone marrow metastases. We agree that it will be interesting to further investigate neutrophils in our pre-neoplastic melanoma fish and other (human) tumor data but believe this is beyond the scope of the current paper.

We added the following paragraph to the discussion:

“However, care must be taken when performing such analysis, as our signatures were designed to identify neutrophil maturation stages and not to distinguish between neutrophils and other cell types. Other cells, in particular closely related myeloid phagocytes, utilize similar genes in their maturation and may influence analysis of bulk tissues. Please refer to Methods for additional assessments of the specificity and sensitivity of the pan-species neutrophil maturation signatures. Therefore, we strongly advise validation of putative differences in neutrophil maturation states using orthogonal assays (e.g., using imaging as illustrated in our study).”

Supplementary Figures 10-15 and applied methods in the section *Cross-species integration and comparison* under Methods.

Supplementary Figure 15: Computational mixing of neutrophil maturation stages and stromal cells. : Specificity of neutrophil maturation signatures in human development atlas. Scatterplots comparing $M1_{pan}$ and $M3_{pan}$ signature scores across pseudo-bulks of human hematopoietic maturation stages from *Suo 2022 et al* across different tissues⁵². Module scores were calculated by *gsva* function from *GSVA* package. The mature neutrophil stage (NEUTROPHIL) shows high $M3_{pan}$ score and low $M1_{pan}$ score, the immature stage (PROMYELOCYTE) shows the opposite pattern, and the intermediate stage (MYELOCYTE) is in-between. In combination, the scores from the two modules resolve and separate the three neutrophil stages across tissues achieving the correct order of maturation (albeit individual scores might be higher, e.g., $M3_{pan}$ for some monocytes).

Minor points:

1. In Fig. 2C, how is linearity calculated? Based on Line 128-130, it seemed that lower linearity represented more straightforward moving pattern, which is confusing.

Linearity of forward progression is a TrackMate feature and is calculated as the ratio between the mean straight line speed and the track mean speed, which is now stated in Material and Methods. The resulting values are similar when calculating the meandering index as track displacement/ total distance traveled. We stated in lines 128-130 that *mmp9+* cells show a lower linearity, indicating a meandering (and not linear) movement in the wound area. Lower linearity does not represent more linear movement, but the opposite.

2. Page 8 Line 190, *cxcr4b* seems up-regulated but not down-regulated in P3. Please modify the description.

Thank you for spotting this mistake! The gene *cxcr4b* is indeed down-regulated in P4 (and not in P3). We have now corrected this.

3. In Fig. 4D, Fig. 5A-C, and Fig. 8, the zebrafish genes were not written in italic.

We have now consistently italicized gene names. Thank you for pointing out this inaccuracy.

Reviewer #2 (Remarks to the Author):

"This is an interesting manuscript that attempts to link MMP9 expression on zebrafish neutrophils with neutrophil maturation. The authors also present a comprehensive comparison of zebrafish, mouse, and human neutrophil transcriptomes. The work represents a significant and rather elegant effort with interesting outcomes. However, I feel that there are some shortcomings in this manuscript that should be addressed.

Thank you for your kind assessment of the significance of our work and for your scholarly critique, which inspired additional analyses which we believe further improved the manuscript.

I am not sure that I agree with the authors' conclusion that *mmp9* expression, as based on the results in Fig.2, correlates with neutrophil maturation. It could be argued that the expression of something like an MMP would be upregulated in situations where a cell type like a neutrophil must extravasate into/migrate through tissues, such as during wound response, infection and/or in the presence of a tumor. There is also a breadth of data suggesting that tumors will elicit MMP expression on myeloid cells in order to facilitate metastasis."

In mammals, neutrophils have been described to synthesize MMP9 during maturation in the bone marrow and to store it in their granules from where it is rapidly released on demand. This is an exception as in other cells MMP9 gets synthesized *de novo* upon stimulation (reviewed in Augoff *et al.* 2022; <https://doi.org/10.3390/cancers14071847>; Rawat *et al.* 2021; <https://doi.org/10.1007/s10555-020-09951-1>). Whether extravasation of neutrophils is dependent on MMP9 *in vivo* is controversial (Bradley *et al.* 2012, <https://doi.org/10.1371/journal.ppat.1002641>; Betsuyaku *et al.* 1998, <https://doi.org/10.1165/ajrcmb.20.6.3558>).

We agree with the reviewer that *mmp9* expression is sensitive to external stimuli such as infections and wounding. For instance, an upregulation of *mmp9* was described for epithelial cells after wounding (LeBert *et al.* 2015, <https://doi.org/10.1242/dev.121160>). Consistently, we also observed an increased frequency of Mmp9+ LysC+ neutrophils (and a higher mean expression of *mmp9*) upon infection with bacteria (**Supplementary Figure 2a, b**). For this reason, we took care to generate our scRNA-seq data from kidney marrow neutrophils under homeostatic conditions. Mmp9 served as beacon for a trajectory of maturation, which was in the next steps supported by modules of numerous genes comprising our neutrophil maturation signature. Through cross-species comparison and focusing on conserved genes (lag<50) the signature became even independent of *mmp9*, which is not included in the final pan signature.

I would argue that neutrophil activation/polarization is not synonymous with differentiation. As an extension of this argument, it is then maybe not adequate to use MMP9 expression as an indicator of neutrophil maturation in their single cell analyses. These results hinge on an unsupported assumption that the level of MM9 expression tracks with maturation, which in my opinion has not been shown in this paper.

In humans, MMP9 is found in tertiary granules (gelatinase granules), which appear during the band cell to segmented cell stage, while lysozyme is already found in primary granules at the myeloblast to promyelocyte stage (Coffelt, Wellenstein & de Visser, 2016, <https://doi.org/10.1038/nrc.2016.52>; Cowland & Borregaard, 2016, <https://doi.org/10.1111/imr.12440>; Cowland, J.B., Borregaard N., 1999, <https://doi.org/10.1111/imr.12440>). This let us to hypothesize that also in zebrafish lysozyme could be used to detect neutrophils from an early developmental stage onwards, while Mmp9 might be more specific to more mature stages.

In our manuscript, we are able to present the following pieces of evidence in support of this hypothesis, which make us confident, that Mmp9+/LysC+ cells are generally more mature neutrophils than Mmp9-/LysC+ cells (under steady state conditions):

- 1) Mmp9+ cells are morphologically more mature than Mmp9- cells: they are smaller cells with segmented nuclei, and have more elongated granules (see cytospin and TEM data in **Figure 1f,g** and **Supplementary Figure 1f,g,i**).
- 2) An unbiased bioinformatics approach based on trajectory inference:
 - a. Placed Mmp9+ neutrophils at the more mature stages along this trajectory, while Mmp9- appeared at early and Mmp9^{intermediate} at intermediate stages (see Figure 3e&f).
 - b. Along maturation pseudotime, *mmp9* expression increases in zebrafish neutrophils and *lyz* expression decreases (see **Figure 3g**).
 - c. Along maturation pseudotime, we observed a switch from proliferative to non-proliferative and anti-apoptotic gene expression, which is consistent with a switch from immature (still proliferative) to mature cell states (see **Figure 4d**).
 - d. Concomitantly, we also detected an increase of known maturation markers like *il1b* and granula components in the mature, Mmp9+ population (**Figure 4d**).
- 3) Complementarily, unsupervised hierarchical clustering of human and mouse neutrophil expression data from previously defined precursor stages, arranged our neutrophil stages in the same order as expected (**Figure 6a**).
- 4) Finally, Mmp9+ neutrophils performed better in classical functions of mature neutrophils like phagocytosis (**Figure 2g**).

“Could the authors demonstrate how MMP9 expression tracks with expression of known markers of neutrophil maturation, such as C/ebp transcription factors, in LysC+ cells? I do not think that it is sufficient to confirm that something like C/ebp is expressed in MMP9+ but not MMP9- cells. Indeed, if as in mammals, C/ebp beta expression is associated with neutrophil maturation, you would expect only matured neutrophils to possess MMP expression. However, if as in mammals, MMP expression is also myeloid cell activation state-dependent, you would also have mature myeloid cells that may not have upregulated expression of a given MMP, depending on their activation, rather than differentiation state. Correlation is not causation, so it could be argued that although the authors see MMP9 expression by cells bearing markers of neutrophil maturation, this does not indicate that MMP9 expression is a reliable marker of neutrophil maturation. Undoubtedly, other populations of fully differentiated fish neutrophils have no/low MMP9 expression.”

Please refer to the plot to the right for the expression of C/EBP family and other TFs along the inferred maturation trajectory (note, those genes are also shown in **Figures 4d and 5a; Supplementary Fig.5**). Compared to published mouse expression data, we see similar dynamics for *hmga1a* and *hmgb2b* with high expression during early maturation phases and for *atf3*, *kif2b*, *junba*, and *cebpb* for late maturation stages. The gene *cebpb1* (ortholog to *Cebpe*) did not show a dynamic expression during the maturation phases captured in our analysis, hence we did not include it in our figures. We have now added the following paragraph in the discussion:

“Another family member, cebpe is known to regulate early neutrophil maturation in mice⁶² and its putative zebrafish orthologue cebp1 has been reported to regulate granule genes lyz and srgn in zebrafish⁶³. Indeed, lyz and srgn were also expressed in early neutrophil maturation in our study (Fig. 4) and cebp1 itself was weakly expressed throughout maturation. Given that there were no strong changes in cebp1 expression during maturation, our algorithms prioritized other well-known hematopoietic regulators such as myb⁶⁴, myca, and ybx1⁴⁷ as driving the early phase of development in our analyses.”

We would argue that biomarkers are correlative in the vast majority of cases, e.g., cell surface markers used to identify hematopoietic cell types by flow cytometry are all not causative for the cell identity, but their expression correlates with the specific cell type. Importantly, we use *mmp9* expression status to sort and enrich cells at the beginning and end of maturation, however, we did not force cells to express certain levels of *mmp9* in our bioinformatics analysis. Therefore, while there is an increase of *mmp9* along pseudotime (see also previous response), the maturation signatures we present in the paper are indeed independent of *mmp9*.

Concerning the reviewer comment that populations of Mmp9 negative mature neutrophils exist, we indeed identified neutrophils without detectable *mmp9* expression at the mature end of our pseudotime maturation trajectory. However, detection uncertainty in scRNA-seq makes it difficult to determine with confidence whether individual mature neutrophil cells without *mmp9* transcript counts are indeed *mmp9*-negative (and they may have already accumulated Mmp9 protein in the granules) or technical artifacts due to inefficient capture of mRNA (“dropouts”). Sparked by the reviewer’s stimulating critique, we have therefore decided to have a closer look at neutrophils in phase 4 (P4) of our maturation trajectory. We re-clustered this subset of cells and looked at *mmp9* expression. The UMAP of this subset shows no clear clustering pattern and cells expressing or lacking *mmp9* are intermingled and mixed all over the UMAP (**Reviewer Figure 2**). That is, cells that have no detected *mmp9* reads are extremely similar to cells with detected *mmp9* based on the expression of thousands of other genes and they do not form a distinct group. We think this shows that cells with/without *mmp9* expression do not present different mature neutrophil states. Rather, lack of detected *mmp9* is likely due to dropouts in the scRNA-seq measurements.

Line 174: It is not clear from the text or the figure legend how the authors are able to deduce the cell cycle stage of the sorted cells. Please clarify.

We apologize for this oversight. Cell cycle phase classification was done based on the expression of G2M and S phase markers using *CellCycleScoring* function from Seurat package. Cells with high level of expression of either G2M or S markers, as they are anti-correlated, are likely to be cycling cells. We now added this information to the figure legend (**Supplementary Fig. 3k** and Methods section *Read processing, quality control, and normalization*).

The legend for Supplementary Figure 3k-m reads now like this:

“**k** UMAP of single-cell RNA-seq data ($n = 18,150$ cells) showing Seurat clustering results using resolution 0.05. The cycling cell cluster is labelled with an asterisk (*). **l** Stacked bar plot showing the percentage of cells per cell-cycle phase in each cluster. The cycling cell cluster is labelled with an asterisk (*). **m** Top 3-enriched gene sets per database (ranked by FDR) from *hyper* overrepresentation analysis of cluster 3 markers.”

It is not clear to me how the analyses of fish neutrophils in Figs. 4-6 relate to the neutrophil populations sorted based on MMP9 expression and depicted in Fig. 3. Please clarify and maybe find a way to indicate that in your figures/legends.

Thank you for pointing out this lack of clarity. Sorting by *Mmp9* expression enriches neutrophils at early and later maturation stages. The distribution of sorted cell populations along maturation pseudotime is shown in **Figure 3f** (previously Figure 3e in the original submission). We attempted to

further clarify this point by adding this distribution to the bottom of the heatmaps (**Figures 4a and 5a**) and additionally improve the respective figure legends. We have additionally added an improved workflow overview in **Figure 3a**.

The cross-species comparisons are very interesting and valuable to the field.

Thank you for your interest in our study!

Reviewer #3 (Remarks to the Author):

“This article generates zebrafish models for different mature stages of neutrophil. Through single cell RNA-seq analysis, the authors discovered four different phases of neutrophil and three gene modules and transcription factor regulators associated with these stages in Zebrafish. By clustering module genes in zebrafish, human, and mouse samples, the expression pattern is similar across species in each stage. These genes also show similar trends along mature stages in zebrafish, human, and mouse. The lag in expression is smaller for early-stage module genes and larger for late-stage module genes across species. Known immature neutrophil gene signatures are more in late-stage gene modules. The resulting gene modules in this article can be an addition to the existing gene signatures. They can differentiate different maturation stages as well as can be a pan-species gene signatures shared among zebrafish, human, and mouse. The application of the signatures to human bulk RNA-seq data shows proper stage estimation of each sample. The discovered gene modules provide a robust alternative to existing neutrophil maturation signature in identifying neutrophil maturation stages from gene expression data. The work is well designed and provides sufficient evidence and explanation.”

We thank the reviewer for the careful reading and positive assessment of our work. The reviewer correctly identified three inaccuracies / not fully worked out points in our initial submission, which we have now attempted to correct (see next responses).

The tissues of the samples used should be explicitly presented. The effects of tissue used on derivation of gene modules and identification of neutrophil maturation stage should be discussed.

Thank you for pointing out that the tissue used was not sufficiently clear. Given that all our data were generated from kidney marrow-derived neutrophils, which was used for defining the zebrafish neutrophil maturation trajectory (first in **Fig. 3**), we also focused on bone marrow neutrophils for all cross-species analysis in **Fig. 6**. We attempted to make this clearer in the main text (line 292) and figure legends (**Fig. 6b-d**). Although we derived signatures based on bone marrow-derived cells only, we found that they were applicable to neutrophils in different tissues – we illustrate this in our response to Reviewer #1, who had raised an interesting question about the specificity of our signatures (**Supplementary figures 10-15**). Briefly, we examined our *M1_pan* and *M3_pan* signatures in single-cell datasets. This included data from different human tissues like bone marrow, blood, lung, or lymph nodes. Our results indicate that in these cases the gene signatures we defined in bone marrow were equally applicable in other tissues.

The authors use slingshot to infer trajectory. As it is prediction method and some of the conclusions are relying on the results of this analysis, it will be more robust to double check with another inference application.

We chose Slingshot because this tool made few assumptions (e.g., allows the inference of the trajectory in a cluster-free manner that only depends on the shape of the data. This has the great advantage of avoiding inference bias based on clustering resolution) and because it was one of the top-performing algorithms in the comprehensive review by the Saeys lab (Saelens *et al.* 2019, <https://doi.org/10.1038/s41587-019-0071-9>). However, we did also try three other popular and well-performing approaches: “first component score” (i.e., simply using the first harmony-corrected principal component), “TSCAN” (Ji et al 2016), and “Monocle” (Cao et al 2019). All approaches yielded high and significant correlation with Slingshot-inferred pseudotime scores (**Supplementary Figure 4**). Moreover, these comparisons highlight the limitations of adopting cluster-based approaches on our data as shown in the scatter plots where some clusters have better concordance than others, and some cells of the same cluster are assigned the same pseudotime score. Overall, we show how different tools generally recover the structure of the trajectory we adopted in our work and the advantage of using Slingshot for our application.

“Transcriptional mechanisms” is mentioned in the title, while few contents are related, thus should be avoided.

We have now replaced “mechanisms” with “signatures”, which we believe is a more truthful representation of the contents of our paper.

New title:

“Cross-species analysis identifies conserved transcriptional signatures of neutrophil maturation”

REVIEWERS' COMMENTS

Reviewer #1 (Remarks to the Author):

The authors have addressed all my concerns. Only one point needs to be addressed for the new data. In the new Fig. S1c, lyz+mpeg1+ double positive cells are more than lyz+mpeg- cells, while in Fig. 1c, the flow cytometry data showed that lyz+mpeg- cells (Q1) percentages seems to be similar with lyz+mpeg1+ cells (Q2) percentages. Is the inconsistency due to different analysis methods?

Reviewer #2 (Remarks to the Author):

I feel that the authors successfully addressed the reviewers' concerns, thus improving an already exciting manuscript.

This work provides important insights that will be valuable to several sub-disciplines.

Reviewer #3 (Remarks to the Author):

The authors addressed the comments.

Response to reviewer comment

We would like to thank reviewer 2 for the additional comment:

“In the new Fig. S1c, $lyz+mpeg1+$ double positive cells are more than $lyz+mpeg-$ cells, while in Fig. 1c, the flow cytometry data showed that $lyz+mpeg-$ cells (Q1) percentages seems to be similar with $lyz+mpeg1+$ cells (Q2) percentages. Is the inconsistency due to different analysis methods?”

We agree with Reviewer 2 that there is a noticeable difference between the two analysis methods, with an overrepresentation of mpeg expressing cells in the confocal data compared to the flow cytometry data. Entire larvae were analysed by flow cytometry whereas only three regions were analysed by confocal microscopy. However, we believe that the observed difference is mainly due to different thresholds for the weak mCherry signal of the mpeg transgene. To make mpeg expression visible and countable by eye for analysis of confocal images the mCherry signal was increased. This might have led to including cells below the mCherry gating threshold of our flow analysis.

To provide some more quantitative data for our confocal image analysis we have now also measured mCherry intensity levels in macrophages ($mpeg+lyz-$) and neutrophils ($mpeg+lyz+$). As expected for a macrophage reporter strain we found that the mCherry signal was 2.5 x higher in macrophages versus $mpeg+$ neutrophils (see Reviewer Figure). This is now mentioned in the results section and reads like this:

“ $Mmp9^+LysC^+$ cells had low $mpeg:mCherry$ expression, while bona-fide macrophages were highly (2.5-fold higher signal) positive for $mpeg:mCherry$ and few of the latter showed detectable $mmp9$ expression (Fig.1b & Supplementary Fig. 1c).”

However, the aim of Fig. 1a-c was to show the distribution of Mmp9 expression in the different myeloid populations, whereas in Supplementary figure 1c we aim to point out the existence and localization of the different Mmp9+ populations within the larvae. We hope that Reviewer 2 agrees with us, that those two points can be safely made by the data we present.

Reviewer Figure: Measurement of $mpeg:mCherry$ signal strength.

Confocal data used for Supplementary Figure 1c was analysed in LAS X using the ROI quantification tool towards mCherry intensity in the different cell populations. Intensities for LysC- Mpeg+ cells (Min-Max: 16-161; mean: 88.6; SD=37.8) and for LysC+ Mpeg+ (Min-Max: 11-114; mean 34.8; SD=20.8) in the CHT and tail were analysed. Background signal was deducted. n = 3 larvae (10 cells/cell type of each larva).